# READOUT REPRESENTATION: REDEFINING NEURAL CODES BY INPUT RECOVERY

**Shunsuke Onoo[1], Yoshihiro Nagano[1, 2], & Yukiyasu Kamitani[1, 2, 3]**
[1]Graduate School of Informatics, Kyoto University, Kyoto, Japan
[2]ATR Computational Neuroscience Laboratories, Kyoto, Japan
[3]Guardian Robot Project, RIKEN, Kyoto, Japan

## ABSTRACT

Sensory representation is typically understood through a hierarchical-causal framework where progressively abstract features are extracted sequentially. However, this causal view fails to explain misrepresentation, a phenomenon better handled by an informational and teleological view based on decodable content and downstream functions. This creates a tension: how does a system that abstracts away details preserve the fine-grained information needed for downstream functions? We propose readout representation to resolve this, defining representation by the information recoverable from features, rather than their causal origin. Empirically, we show that inputs can be accurately reconstructed even from heavily perturbed mid-level features, demonstrating that a single input corresponds to a broad, redundant region of feature space, challenging the causal mapping perspective. To quantify this property, we introduce representation size, a metric linked to model robustness and representational redundancy. Our framework offers a new lens for analyzing how both biological and artificial neural systems learn complex features while maintaining robust, information-rich representations of the world.

## 1 INTRODUCTION

The dominant view of neural sensory representation is rooted in a hierarchical-causal framework, where a stimulus is processed through layers that extract progressively abstract features, and the causal origin is treated as the contents of representation (DiCarlo et al., 2012; Kriegeskorte et al., 2008). This model, aligned with causal theories of representation in philosophy (Fodor, 1987), has spurred powerful analytic tools (Kornblith et al., 2019; Seung & Lee, 2000). However, its strict causal foundation fails to explain misrepresentation—a phenomenon like illusions or mental imagery, where representational content is decoupled from a direct sensory cause (Fodor, 1987). Alternative philosophical theories address this gap: informational accounts define representation by the information that a state carries (Dretske, 1981), while teleological accounts define it by its proper function for a downstream consumer (Millikan, 1989). These non-causal views are empirically supported by neural decoding studies, which show that subjective content like dreams and imagery can be read out from brain activity using decoders trained with stimulus-induced perception (Horikawa et al., 2013; Cheng et al., 2023; Kamitani et al., 2025).

These competing perspectives create a central tension: how can a system designed for hierarchical abstraction, which supposedly discards details, simultaneously preserve the fine-grained information required for downstream functions? Both theoretical and empirical work has shown that detailed information remains recoverable even from higher-level representations with large receptive fields (Zhang & Sejnowski, 1999; Majima et al., 2017). In deep neural networks, fine-grained visual details can be reconstructed from upper layers (Mahendran & Vedaldi, 2015). These findings confirm that abstraction and detail retention are not mutually exclusive, suggesting that population codes can support both simultaneously.

To formally reconcile these observations, we introduce *readout representation*, a framework that operationalizes insights from informational and teleological theories. We define a representation not by its causal origin, but as the set of all neural features from which a specific signal can be functionally recovered (Figure 1A). This approach provides a concrete, quantifiable method for

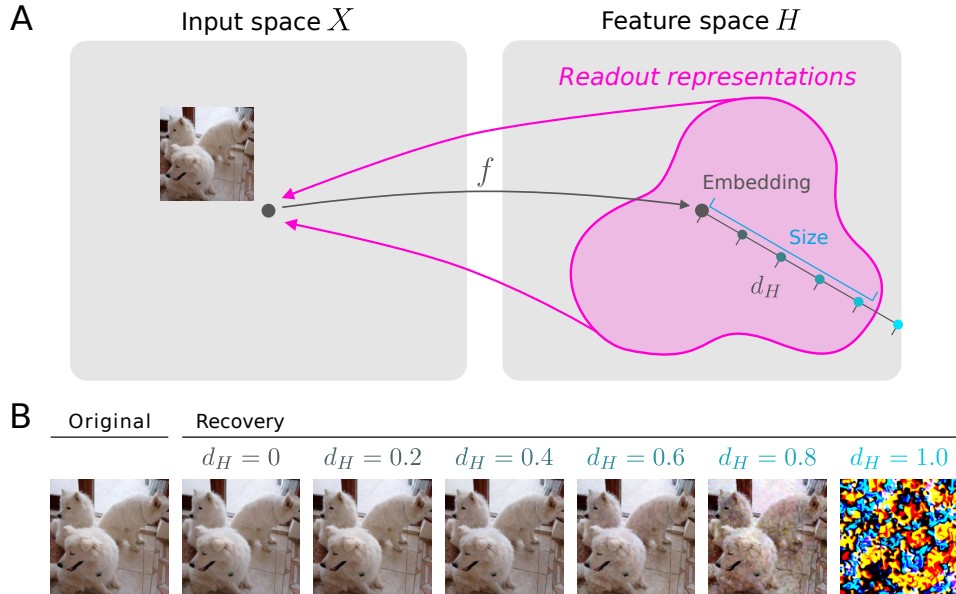

Figure 1: Concept of readout representation and images reconstructed from perturbed features. **A**: Concept of readout representation. Traditionally, a neural representation is defined by a causal relationship—by the input that elicits representation. In contrast, we propose to define representations by the information recoverable from neural features. Under this framework, the representation of a single input can be a region in feature space, consisting of all points from which the input information can be read out. **B**: Images recovered from perturbed features in conv3_1 layer of VGG19. From left to right, the original image, the image recovered from the original feature, and the images recovered from perturbed features. Above the recovered images, $d_H$ indicates the correlation distance between the original feature and the perturbed features. Input images can be recovered from heavily perturbed features, showing that an input image is represented by a broad region in representational space, supporting the readout view. We provide full results in Appendix E and Appendix H.

analyzing how information is preserved throughout a system's hierarchy, moving beyond abstract philosophical distinctions. It offers a practical lens for studying how both biological and artificial systems learn abstract features while maintaining robust, information-rich codes.

To explore our framework's implications, we used deep neural networks as a fully observable testbed. Our investigation revealed a striking phenomenon: a single input corresponds to a vast and continuous region of feature space from which it can be recovered. To systematically characterize this discovery, we probed the boundaries of these readout representations by applying feature inversion (Mahendran & Vedaldi, 2015) to deliberately perturbed features. We found these representational sets to be surprisingly large across both vision and language models, with inputs remaining recoverable even from features displaced far from their canonical values (Figure 1B). To quantify this property, we introduce *representation size*, a metric that captures this robustness and correlates with representational redundancy and model performance. Experiments with simplified models further confirm that this is a general principle of neural representation, not an artifact of deep networks.

Our key contributions are as follows:

- **Conceptual framework**: We introduce *readout representation*, a framework that defines representation by functional recoverability rather than causal origin, resolving the tension between hierarchical processing and information preservation.

- **Empirical validation**: We demonstrate across diverse models that input information is recoverable from a wide range of perturbed features, establishing the generality of our framework.

- **Quantitative measure**: We propose *representation size* as a novel metric that captures the extent of this recoverable feature space, linking it to representational redundancy and model robustness.

## 2 RELATED WORK

**Philosophical theories of representation.** Naive accounts, called crude causal theory, define mental representations by their causal origin, where a state represents the causal origin that produced them (Fodor, 1987). This view struggles with misrepresentation, such as illusions or imagery, where representational content diverges from external causes. Informational theories instead define representation by the information carried by a state (Dretske, 1981), while teleosemantic accounts emphasize proper function for downstream consumers (Millikan, 1989). Our work operationalizes these non-causal perspectives by defining representation in terms of recoverability.

**Hierarchical-causal views in neuroscience and machine learning.** Much empirical work interprets neural representations through progressive abstraction of input stimuli. This has motivated concepts such as neural manifolds (Seung & Lee, 2000; DiCarlo et al., 2012; Poole et al., 2016; Sorscher et al., 2022; Chung et al., 2018), representational similarity analysis (Kriegeskorte et al., 2008; Kornblith et al., 2019; Huh et al., 2024), and information bottleneck theory (Tishby & Zaslavsky, 2015; Shwartz-Ziv & Tishby, 2017). Hierarchical receptive-field analyses in deep networks further reveal preferred features and visualization methods (Zeiler & Fergus, 2014; Simonyan et al., 2014). These approaches treat features as encodings of their causal inputs, whereas we shift focus to the information that can be read out from features irrespective of origin. Note that our aim is not to reject the hierarchical-causal view, but rather to reconcile the tension between hierarchical abstraction and retention of fine-grained information, grounded in the informational and teleological account of neural representation.

**Recoverability-based approaches.** A complementary line of work investigates representation by the information that can be recovered from neural features. Feature inversion uses gradient-based optimization to recover inputs (Mahendran & Vedaldi, 2015), and probing uses linear classifiers to quantify decodable content (Alain & Bengio, 2017). Brain decoding and reconstruction studies have shown that diverse information and perceptual experiences can be read out from neural activity (Kamitani & Tong, 2005; Miyawaki et al., 2008; Horikawa & Kamitani, 2017a; Shen et al., 2019; Findling et al., 2025; Sucholutsky & Griffiths, 2023), including those decoupled from causal origin such as attended stimuli (Kamitani & Tong, 2005; Horikawa & Kamitani, 2022), visual illusions (Cheng et al., 2023), mental imagery (Shen et al., 2019), and dreams (Horikawa et al., 2013; Horikawa & Kamitani, 2017b). Leveraging the feature inversion technique, Feather et al. (2023) investigated *metamers*, which are multiple distinct inputs that converge to identical representations, revealing the model's inability to distinguish them. Conversely, our readout representation framework captures cases where multiple neural features represent a single input, characterizing the robustness of the model's representation. Building up on these studies, we propose readout representation as a computationally tractable formalization of representation. This formulation differs from previous work by providing a rigorous treatment of misrepresentation and accommodating cases in which multiple neural features represent a single information. This formalization also allows us to examine the size of such representations, providing a new lens to reconcile the tension.

**Redefining neural representations.** Studies have proposed frameworks for redefining neural representations: by the information that drives behavior (Panzeri et al., 2017), by their informational properties (Pohl et al., 2025), or from a philosophical perspective (Churchland & Sejnowski, 1990). Panzeri et al. (2017) focus specifically on behavior, while ours incorporates arbitrary readout procedures not limited to behavior. Unlike these studies, our framework offers a computationally tractable formalization of representation, a coherent handling of representations decoupled from causal origin, such as misrepresentation and dreaming, and a quantitative metric to characterize the robustness of representations.

**Redundancy and robustness in neural representations.** Studies have proposed metrics to characterize redundancy in neural representation through mutual information and partial information decomposition (PID) (Schneidman et al., 2003; Williams & Beer, 2010), and through compression (Tishby & Zaslavsky, 2015; Shwartz-Ziv & Tishby, 2017). These approaches analyze redundancy and compression at the dataset level, whereas our representation size quantifies redundancy at the

instance level, offering new insights into how individual inputs are robustly encoded. In terms of robustness, adversarial examples demonstrate that small perturbations can drastically alter encodings (Szegedy et al., 2014), and this sensitivity is attributed to the presence of features rather than artifacts (Ilyas et al., 2019). Probabilistic generative models exemplified by VAEs (Kingma & Welling, 2014; Rezende et al., 2014) explicitly introduce variability into the representations of artificial neural networks. While these studies offer insights into how to incorporate variability into representations, our work focuses on characterizing the variability inherently present even in deterministic models trained for standard tasks.

## 3 READOUT REPRESENTATION

We introduce the readout representation framework, which formalizes neural representation by the information that can be recovered and used for downstream functions. Consider the brain or neural network $f : X \times \Xi \to H$ that maps an input stimulus $x \in X$ to a neural feature $h \in H$ under a brain state or context $\xi \in \Xi$. Let $S$ denote a signal space of interest, and let the true signal corresponding to the input $x$ be given by a reference mapping $\bar{\pi} : X \to S$. The signal space $S$ can be the input itself, a compressed description, or any latent variable of interest. Let $\pi : H \to S$ denote a readout procedure, which extracts signal from features. We assume the spaces $H$ and $S$ are each endowed with equivalence relations.

Under the causal view, a feature $h = f(x)$ is considered a representation of the input $x$ that causally generated it. When there is stochasticity between $x$ and $h$, such as in brain activity or VAEs, one typically considers the expectation $\mathbb{E}_\xi[f(x, \xi)]$. In contrast, the readout representation treats the feature $h$ as a representation of $s$ based on the information it conveys (See Appendix C for the treatment of randomness in this framework).

### 3.1 DEFINITION OF READOUT REPRESENTATION

Following the informational view, representation is defined by the information recoverable from the feature using a readout method $\pi$, independently of its causal origin.

**Definition 1** (Representation). *$h \in H$ represents $s \in S \iff s = \pi(h)$.*

This definition contrasts with the approach adopted by the hierarchical-causal view, which defines the feature $h$ as a representation of its causal origin $x$. Under this definition, multiple features may represent the same signal; we term the set of all such features *readout representations*.

**Definition 2** (Readout representation). *The set $H_s^\pi := \{h \in H \mid \pi(h) = s\}$ denotes readout representations of a signal $s$.*

The readout representation $H_s^\pi$ depends on the choice of the readout procedure $\pi$, and the interpretation changes accordingly. For practical applications, careful consideration must be given to the permissible class of readouts according to the problem being addressed. See Appendix A for further discussion.

Given the set $H_s^\pi$, we can evaluate the spread of the set $H_s^\pi$ as *representation size*, which can be interpreted as the volume occupied by specific information within the feature space. The representation size can be instantiated in various ways, such as cardinality, geometric volume, or other distance-based or probabilistic indicators.

Note that the readout representation is a natural extension of the causal representation when the readout is the inverse of the encoding function.

**Remark 3.** *Suppose $S := X$ and $f$ is injective, and ignore the context $\xi$. Define $\pi(h) := f^{-1}(h)$ for $h \in \text{Im } f$, and define $\pi(h)$ to be an arbitrary element of $X$ otherwise. Then the causal representation $h = f(x)$ lies in the readout representation $H_x^\pi$. Even in this case, any other feature $h' \neq h$ is also included in $H_x^\pi$ as long as $\pi(h') = \pi(h)$.*

Readout representation enables us to handle *misrepresentation* as a situation where the extracted signal differs from the corresponding true signal. To exclude trivial cases where all features represent the same signal, we first introduce *representation capability* as the ability of $(f, \pi)$ to distinguish signals.

**Definition 4** (Representation capability). $(f, \pi)$ *has representation capability of $S$ if and only if*

$$\exists x_1, x_2 \in X, \exists \xi \in \Xi \quad s.t. \quad \bar{\pi}(x_1) \neq \bar{\pi}(x_2), \bar{\pi}(x_1) = \pi \circ f(x_1, \xi), \bar{\pi}(x_2) = \pi \circ f(x_2, \xi).$$

**Definition 5** (Misrepresentation). *Suppose $(f, \pi)$ has representation capability of $S$, and feature $h = f(x)$ represents $s = \pi(h)$. Then, we say $h$ misrepresents $s$ if and only if $\pi(h) \neq \bar{\pi}(x)$.*

## 3.2 CASE STUDIES

Readout representation offers a formal treatment of neural representation decoupled from its causal origin, which is challenging to accommodate within the causal view (See Appendix B for a detailed discussion). In the following case studies, we demonstrate how readout representation accommodates misrepresentation, illusion, and dreaming.

**Misclassification.** Let the input space $X$ be natural scenes, the feature space $H$ be neural features in the visual cortex, the signal space $S$ be object categories of attended objects, and the reference mapping $\bar{\pi}$ assigns the true category. Suppose a subject sees a scene $x \in X$ that contains a rope but mistakenly perceives it as a snake. First, we consider $\pi_o : H \to S$ as the subject's oral report to illustrate how the framework treats misclassification. The subject's neural feature $h = f(x, \xi)$ represents the signal "snake" since $\pi_o(h) = $ "snake". In our framework, the feature $h$ misrepresents the input, as $\pi_o(h) = $ "snake" $\neq \bar{\pi}(x) = $ "rope". The framework also covers a case where the subject views the scene again under a different state $\xi'$ and reports "rope" as $\pi_o(f(x, \xi')) = $ "rope". Second, instead of oral report, we consider brain decoder $\pi_d$ that recovers visual information from neural features in each region (ROI). By applying $\pi_d$ to each region, we can compare the decoded content across the hierarchical processing. The decoded content from early visual regions would be closer to "rope", whereas the one from higher regions would be closer to the categorical content "snake." By identifying the transition between regions that accurately represent the stimulus and those that misrepresent it, the framework allows us to pinpoint where misrepresentation emerges and potentially analyze the mechanism behind it. This formalization aligns with previous decoding studies that analyze neural representations decoupled from a causal origin (Cheng et al., 2023; Kamitani et al., 2025).

**Illusion.** Consider a Müller–Lyer figure, where two parallel lines of equal length appear different in length. Let $X$ be images that contain two parallel lines, $d(x)$ be the difference in length of two lines, and $S$ indicate whether this difference exceeds a perceptual threshold $\Delta$. Let the reference mapping be $\bar{\pi}(x) = \mathbb{1}_{d(x) > \Delta}$, and the readout $\pi_o$ be the subject's oral report. For ordinary images, the subject correctly classifies differences, so the representation capability is satisfied. For the illusion image $x$, there are no differences in length, so $\bar{\pi}(x) = 0$. However, the subject may report $\pi_o \circ f(x, \xi) = 1$. Within our framework, the feature $h = f(x, \xi)$ misrepresents the length difference, as $\pi_o(h) = 1 \neq \bar{\pi}(x) = 0$.

**Dreaming.** During sleep, neural activity can generate representations without external input. Let $X$ be visual stimuli and $S := X$. Let $\pi_o$ be the subject's oral report. Under wakefulness ($\xi_w$), the subject can describe stimuli through oral reports, establishing representation capability. Under sleep ($\xi_s$), if the subject dreams of a dog, then $\pi_o \circ f(x, \xi_s) = $ "dog". Here the feature $h = f(x, \xi_s)$ represents content decoupled from the environment. Similarly, consider a brain decoder $\pi_d$ trained on wakeful activity as in (Horikawa & Kamitani, 2017b). The decoder may output "dog" when applied to the sleep activity, again producing a valid representation even without external input. The framework thus treats dream content as part of the representational space defined by readout, integrating it into the same formal structure that applies to waking perception.

## 4 EXPERIMENTS

We examine representations of deep neural networks through the lens of readout representation. Specifically, we demonstrate that, contrary to the idea that information is progressively discarded, crucial input details are robustly preserved across a broad range of hierarchical features. First, we show that information about an input stimulus is often represented in a broad region of the feature space across multiple models, modalities, and layers. We show this by demonstrating that the input information is recoverable from heavily perturbed features. Second, we show the potential of the representation size metric to analyze representational properties of different input stimuli, model

architecture, and model size. Finally, we experiment with a simplified model to highlight the role of redundancy in the robust recovery from perturbed features.

The following experimental settings were applied across experiments unless otherwise specified. We provide details in Appendix D.

**Feature inversion.** We instantiate the readout $\pi$ with feature inversion because it makes minimal structural assumptions about $\pi$ and searches the preimage of $f$ directly. If input-level information is present in the feature, inversion will retrieve it without relying on task-specific decoders. Specifically, given a target feature $h$, we define $\pi(h) = \arg\min_x L(f(x), h)$ where $L$ is a feature-matching loss. For vision models, we used Deep Image Prior (DIP) (Ulyanov et al., 2020) as a prior to reduce high-frequency artifacts while avoiding external pretrained generative priors. We also examined the robustness of our findings without DIP in Appendix G. For language models, we optimized token logits, which are converted to embeddings and fed into the model while iteratively minimizing the distance to the target features.

**Distance and accuracy measures.** We used correlation distance[1] as a distance function in the feature space $d_H$ across both modalities, and as a distance function in the input space $d_X$ for the vision modality because it yields a unit-free, dimension-independent scale that enables comparisons across layers and modalities. For the language modality, we used token error rate as a distance measure in the input space $d_X$, defined as the proportion of tokens in the reconstructed text that differ from the original text. We verified the robustness to the choice of $d_X$ in the vision modality by repeating evaluations with perceptual pixel-space metrics (SSIM (Wang et al., 2004), PSNR, LPIPS (Zhang et al., 2018), and DISTS (Ding et al., 2022)) (Appendix E).

**Readout representation.** To account for minor deviations due to numerical inaccuracies in the optimization process, we used a relaxed version of readout representations parameterized by a threshold $t \geq 0$ on input-space distances: $H_{x,t}^\pi = \{h \in H \mid \forall x' \in \pi(h), d_X(x, x') < t\}$. The threshold $t$ was chosen as a sufficiently small value specific to the modality and distance measure: 0.1 correlation distance in the vision modality, and 0.3 token error rate in the language modality. We instantiate the representation size by the maximum feature-space deviation within the relaxed set: $r_x = \max\{d_H(h, f(x)) \mid h \in H_{x,t}^\pi\}$.

**Feature perturbation.** As a feature perturbation, we added Gaussian noise to the original feature across 10 noise levels, calibrated to produce specific correlation distances from the original feature.

### 4.1 FEATURE INVERSION FROM PERTURBED FEATURES

We first examined how broadly an input is represented in the feature space of deep neural networks by reconstructing inputs from perturbed features. We conducted experiments across both the vision and language modalities. For the vision modality, we used VGG19 (Simonyan & Zisserman, 2015), CLIP (Radford et al., 2021), and DINOv2 (Oquab et al., 2024), which span both convolutional neural networks (CNNs) and Vision Transformers (ViTs). We also performed experiments using a variational autoencoder and report results in Appendix E. For language, we used BERT (Devlin et al., 2019), GPT2-series (Radford et al., 2019), and OPT-series (Zhang et al., 2022). In both modalities, we performed reconstruction using 64 samples: randomly sampled images from ImageNet (Deng et al., 2009) and texts from C4 (Raffel et al., 2020). Note that our aim is to illustrate how multiple features can represent a single input, rather than to benchmark model performance. We examined features in multiple layers for each model: all convolutional layers in VGG19 and every quarter-depth transformer block in Transformer models.

We found that the original images can be reliably reconstructed even from significantly perturbed features in lower to middle layers in multiple vision models (Figure 1B). For example, in the lower to middle convolutional layers of VGG19, near-perfect recovery was observed at feature correlation distances up to 0.8. Quantitatively, in VGG19 lower layers, features perturbed up to 0.7 correlation distance still yielded reconstructions within 0.1 correlation distance in pixel space (Figure 2). Similarly, high-fidelity recovery from perturbed features was also observed in the lower to middle layer features in DINOv2 and CLIP models. Full results, including reconstructed images from all models, are provided in Appendix E. We also evaluated the robustness of our findings against the choice of

---

[1]Although correlation distance (defined as $1 -$ correlation coefficient) does not lead to a proper distance metric, it suffices for our purpose of ranking similarity consistently across heterogeneous spaces.

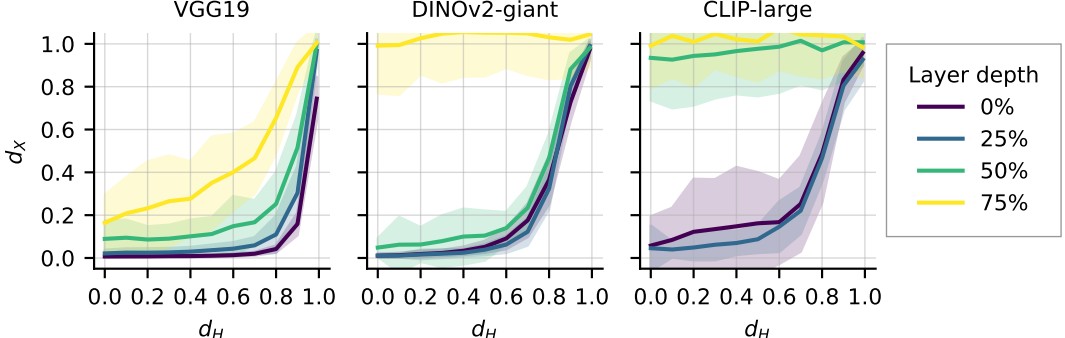

Figure 2: Images can be accurately recovered from heavily perturbed features. The x-axis shows the correlation distance between the original and perturbed features, and the y-axis shows the correlation distance between the original and reconstructed images. Each line represents a quarter-depth layer, colored by layer depth. Shaded areas indicate $\pm 1$ SD across images: for each distance bin, we computed the standard deviation across images. **Left**: In VGG19, lower to middle layers retained high-fidelity information (within 0.1 correlation distance) even under 0.7 correlation distance perturbation. **Middle** and **Right**: Similar patterns were observed in lower to middle layers of DINOv2 and lower layers of CLIP. Full results are provided in Appendix E.

the prior in the feature inversion. While the use of DIP improved reconstruction quality, we observed similar qualitative trends even without DIP (Appendix G). This indicates that the extensive readout representations are not an artifact of the prior, but rather an inherent property of the learned features. We also confirmed that the same quantitative trends were observed when using other measures such as SSIM, PSNR, LPIPS, and DISTS for the input space distance (Appendix E).

A similar pattern was observed in some of the language models (Figure 3). Across models, input tokens were recovered from perturbed features with accuracy significantly above the chance level. In particular, BERT and OPT-350m showed near-perfect recovery even at high perturbation levels (up to 0.7 correlation distance) in lower layers. GPT2 and some OPT variants exhibited reduced performance, but still exceeded chance at certain perturbation levels. Note that, given the large vocabulary sizes (BERT: 30,522; GPT2: 50,257; OPT-350m: 50,272), random guessing yields near-zero accuracy. Additional results, including comparisons across model sizes, recovered sentences, and analyses of model differences, are provided in Appendix H.

Together, these results illustrate that the representations of identical or nearly identical stimuli often extend to a broad region in the feature space, a property observed across multiple architectures, modalities, and layers. Such extensive representational coverage suggests inherent redundancy in neural representations, which enables neural networks to simultaneously retain fine-grained information and achieve hierarchical abstraction.

## 4.2 APPLICATION OF REPRESENTATION SIZE TO INSTANCE-LEVEL ANALYSIS

Next, we examined the utility of representation size for analyzing the neural representations of different input instances. As a case study, we compared representation sizes across different image types using VGG19, focusing on their relationship with model performance. First, we compared images that were correctly classified by the model ("hit") and those that were misclassified ("miss"). We collected 8 hit images and 8 miss images from the ImageNet validation set, where hit images were defined by correct top-1 prediction and miss images were defined by failure to include the correct label in the top-5 predictions. Hit images consistently exhibited larger representation sizes, particularly in higher layers (Figure 4, left). Second, we compared the representation sizes between natural images and noise images generated by sampling pixel values from a uniform distribution. Noise images exhibited zero representation sizes across all layers, in contrast to the broader representations seen for natural inputs (Figure 4, right).

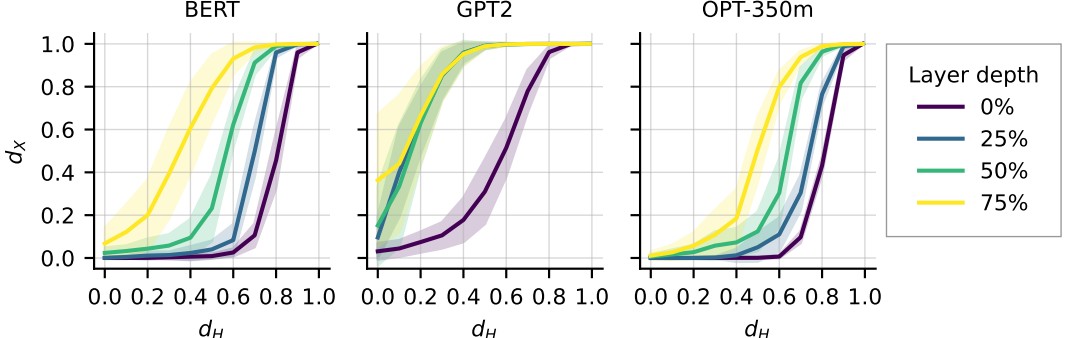

Figure 3: Input texts can often be recovered with high accuracy from perturbed features. The x-axis shows the correlation distance between the original and perturbed features, and the y-axis shows the token error rate. Each line shows results for features from a quarter-depth layer, colored by layer depth. Shaded regions indicate $\pm 1$ SD across samples. **Left**: Similar to the vision modality, the input tokens are recovered with high accuracy even from heavily perturbed features in lower to middle layers. **Middle**: In GPT2, we do not observe high-fidelity recovery from perturbed features, although the accuracy was substantially above chance level across most feature distances. **Right**: Lower to middle layers of OPT-350m and BERT show extended readout representations. We provide further results in Appendix H.

These results suggest that the representation sizes are related to the nature of the instance and whether the model is able to represent it effectively, and reflect the model's perceptual reliability. We performed an additional analysis on VGG19 with randomly initialized weights to examine if these differences in representation sizes were attributable to the distribution of training data (Appendix F). We found similar differences in the VGG19 with random weights, revealing that the differences in representation sizes reflect the bias of the model architecture towards certain types of images. This highlights the potential effectiveness of the representation size for analyzing model representations for specific instances. This analysis differs fundamentally from previous methods that analyze a set of input instances, such as manifold analysis (Seung & Lee, 2000), representational similarity analysis (RSA) (Kriegeskorte et al., 2008), or quantifying the amount of information such as mutual information, as representation size enables the analysis of a single instance.

### 4.3 INTERPRETATION OF READOUT REPRESENTATION

What factors underlie the extended readout representations? We examined the relationship between representation sizes and feature dimensionality, and used a simplified toy model to illustrate how high-dimensional mappings naturally yield redundant, robust representations that support extended readout representations.

First, we examined the relationship between feature dimensionality and representation size across layers of various vision models (Figure 5). Overall, we observed a trend that layers with higher feature dimensionality exhibit larger representation sizes, although the trend weakened in the higher layers. This result suggests that redundancy enabled by high-dimensional representations is a key factor in representation size, while deeper layers may trade this redundancy for compactness and task-specific abstraction. The relationship between feature dimensionality and representation size points to connections with prior work on intrinsic dimensionality. These studies have demonstrated that natural data (Pope et al., 2021) and parameter optimization trajectories (Aghajanyan et al., 2021) possess intrinsic dimensionality much lower than their ambient dimensionality, suggesting that multiple features may represent the same information when the ambient dimensionality is high. The smaller size of representations in higher layers may be influenced by the characteristics of the labeled data and the objective function used in model training. This motivates the toy model analysis, which isolates the role of redundancy under controlled conditions.

Second, we examined the readout representation of a simplified toy model to illustrate the emergence of extended readout representations. The model consisted of 100 neurons with bell-shaped tuning

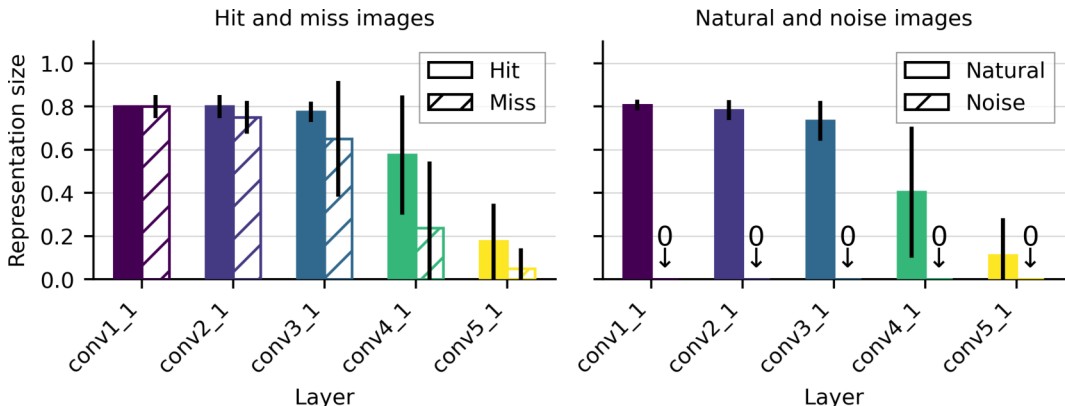

Figure 4: Comparison of representation sizes across different image types using VGG19. Each bar shows the mean representation sizes across images for each image type, and error bars show $\pm 1$ SD across samples. **Left**: Correctly classified images ("hit") exhibit larger representation sizes than misclassified ones ("miss"), particularly in deeper layers. This suggests that successful classification is associated with more redundant and robust representations. **Right**: Comparison between natural and noise images. Noise images show zero representation sizes across layers. We performed an additional analysis and found that these differences in sizes reflect the bias of the model architecture towards certain types of images (Appendix F).

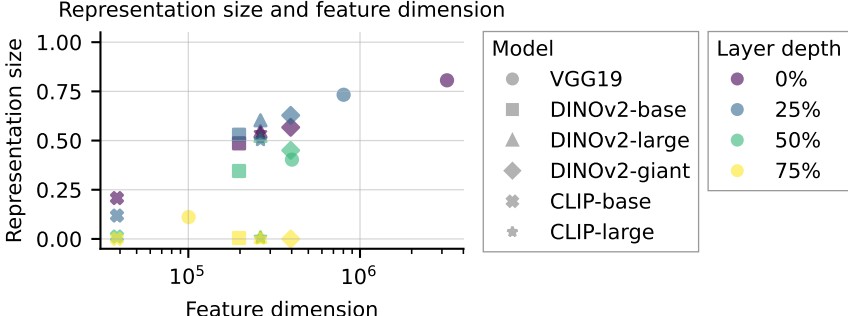

Figure 5: Relationship between representation size and feature dimension across models and layers. Each point represents a layer from each model, plotted by its feature dimensionality (x-axis, log scale) and mean representation sizes across images (y-axis). Color indicates relative layer depth, and marker shape denotes the model. We observed a trend that larger dimensionality leads to larger representation sizes, suggesting that redundancy enables robust representations that support extended readout representations.

curves distributed evenly across a one-dimensional input space $X = [0, 1)$ (Figure 6, top left). Each stimulus $x \in X$ was projected onto a higher-dimensional neural feature space $h = f(x)$. Since the dimensionality of the feature space (100) is higher than that of the input space (1), there is a redundancy in representation, and the features form a low-dimensional manifold (Figure 6, bottom left). This redundancy would allow robust recovery: the original input would be recovered from perturbed features. To test this prediction, we perturbed the feature vectors by adding noise and attempted to recover the original inputs by retrieving the closest features to the perturbed points. In the principal component (PC) space, we projected the points from which we could read out the original input (Figure 6, right). Projected points were distributed in a broad region of the PC space, showing that multiple features represent a single input. Although this toy model is simplified, a similar mechanism would underlie readout representations in deep neural networks, given the low intrinsic dimensionality of input spaces relative to their feature spaces (Pope et al., 2021).

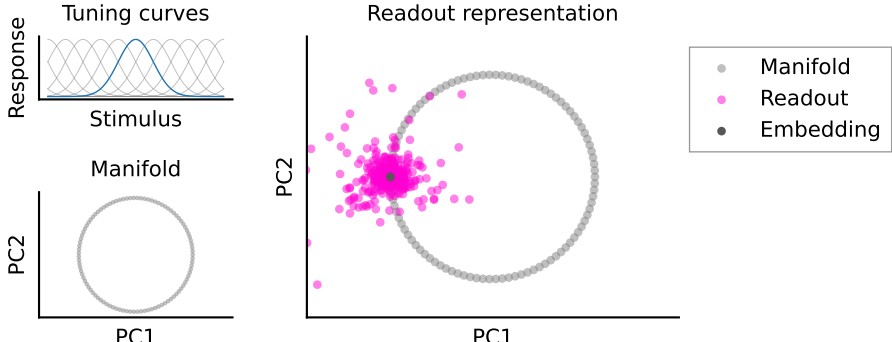

Figure 6: Readout representations in a toy model with bell-shaped tuning curves. **Top left**: Tuning curves of 10 neurons out of 100 neurons, each selectively responding to different locations in a one-dimensional stimulus space. **Bottom left**: The resulting neural manifold, visuzlized by projecting the neural features onto the first two principal components. **Right**: The readout representations of a single input (a dark gray dot) visualized by magenta points. Each readout representation point shows a perturbed feature from which the original input was successfully read out, showing that multiple features represent a single input.

## 5 DISCUSSION

Our study introduced *readout representation*, a framework that defines representations by the information recoverable from features rather than their causal origin. This perspective helps resolve the tension between hierarchical abstraction and detail retention: while abstraction emphasizes categorical or task-relevant features, our results show that fine-grained input information remains recoverable across broad regions of feature space. Through experiments in both vision and language models, we demonstrated that inputs can be reconstructed from heavily perturbed features, indicating that representations are not single points but extended sets. To quantify this property, we proposed *representation size*, which measures the extent of regions that support accurate recovery and links redundancy to robustness and performance. The toy model demonstrates how high-dimensional embeddings produce extended readout representations, providing a mechanistic explanation for the empirical trends. Together, these findings show that neural representation leverages redundancy to achieve both hierarchical abstraction and robust retention of input details, challenging the notion that abstraction necessarily entails information loss. In relation to our original motivation: (i) we quantified recoverability via representation size, (ii) we linked it to performance and redundancy, and (iii) a simplified model accounted for the possible mechanism on how representations can retain recoverable detail while supporting hierarchical abstraction.

This work offers several future directions. First, the representation size can be a tool for analyzing neural representations in deep neural networks. The representation size offers a per-sample analysis of representational redundancy and robustness, complementing existing methods that typically analyze representations at the dataset level (Schneidman et al., 2003; Denil et al., 2013; Feather et al., 2023; Kriegeskorte et al., 2008; Kornblith et al., 2019; Huh et al., 2024). Our experiments show that representation size is related to the accuracy of a model, indicating the potential for diagnosing model failures or understanding why particular examples are difficult for a network. For example, in a machine learning setting, one could compute the representation size of a specific input that exhibits unexpected behavior and compare it with those of typical inputs: smaller sizes could serve as signals for outlier detection, confidence estimation, or targeted debugging. It can also be used to test hypotheses about neural representation; one example is to test whether the representation size reflects the probability of a sample under a prior distribution implicitly induced by the training data. Another direction is to apply the readout representation to biological neural networks as a measure of redundancy and robustness of neural representations, although we note that the method of using perturbed features may not be directly applicable to the brain. Additionally, brain decoding studies often predict deep network features from noisy brain activity (Shen et al., 2019; Cheng et al., 2023), and their success may be explained by the large representation sizes of the networks employed. Building models with even larger representation sizes could improve decoding accuracy.

## REPRODUCIBILITY STATEMENT

Experimental details, including model configurations and hyperparameters, are provided in Appendix D. For experiments involving stochasticity, random seeds were fixed to ensure reproducible outcomes. Source code is provided in the supplementary material.

## ACKNOWLEDGMENTS

We thank our laboratory members for their valuable feedback. This work was supported by Japan Society for the Promotion of Science (JSPS: KAKENHI grants JP25H00450 to Y.K., and JP21K17821 to Y.N.), Japan Science and Technology Agency (JST: CREST grants JPMJCR22P3 to Y.K.), and Guardian Robot Project, RIKEN.

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

APPENDIX

We provide additional information of our study. Appendix A describes the limitations of our work. Appendix B describes the relationship between the causal view and misrepresentation. Appendix C discusses the treatment of noise in our framework. Appendix D describes the details of the experiment including procedures for feature inversion, feature perturbation, optimization settings, and model specifications. Appendix E provides additional results for vision models, including reconstructed images and quantitative results. Appendix F provides additional results for the application of representation size to instance-level analysis. Appendix G provides the ablation results without Deep Image Prior (DIP) in vision experiments. Appendix H reports the additional results for language models, including reconstructed sentences, quantitative evaluations, and insights into the difference between models. Appendix I provides the broader impact of our study. Appendix J describes the usage of LLMs in our study.

## A    LIMITATIONS

Despite our contributions, several limitations remain. First, since the readout representation depends on the choice of readout procedure $\pi$, both the selection of $\pi$ and the interpretation of the results require careful consideration. Specifically, if a strong prior, such as diffusion models, is used as part of the readout procedure, it can unfaithfully inflate the recoverable information, as pointed out in Shirakawa et al. (2025). In our experiments, we used Deep Image Prior (DIP) as a prior in the vision experiments, and we did not use any prior in language experiments. We choose DIP because it is a weak, data-independent prior. Strong generative priors such as GANs or diffusion models could yield even smoother reconstructions, but we intentionally avoided them because they impose powerful biases from the training data distribution of these generative models. By contrast, DIP is widely regarded as a substantially weaker prior: it does not require training and therefore is not biased by specific training data. Importantly, even without DIP (Appendix G), the recovered images remain visibly similar to the original inputs under substantial perturbations of the features. Although we observed relatively sharp increase in the input distance to the small feature perturbations, it reflects high-frequency variations rather than the loss of essential information for humans and machine learning models to perform reasonably natural downstream tasks. This behavior would not be expected if the recovery were solely an artifact introduced by DIP.

Second, the source of variation in representation size across architectures remains unclear. For example, language models such as GPT2-Large and OPT-1.3b showed lower recovery performance under perturbation compared to others (Appendix H). In vision models, we found that feature dimensionality has a strong effect on representation size, and we also provide additional analysis of performance differences among language models in the Appendix H.3, though a comprehensive understanding remains an open question.

Third, while we propose representation size as a metric for model evaluation, our experiments demonstrate only preliminary use cases. Its broader utility remains to be established.

Fourth, although our framework is inspired by both artificial and biological systems, its application to biological neural networks remains an open question. In our study, we used perturbed features to probe the feature space in deep neural networks, but this method is not directly applicable to the brain.

Finally, our implementation adopts feature inversion as the readout method, which introduces some computational overhead. However, our proposed framework is independent from the choice of readout method, and users may adopt alternative procedures better suited to their needs and computational budgets.

## B    CAUSAL VIEW OF REPRESENTATION AND MISREPRESENTATION

The causal view refers to a view that defines representation as follows: a state $h$ represents stimulus $x$ if and only if $h$ is caused by $x$. Fodor (1987) named this view of representation the crude causal theory because the content of representation is derived from the causal origin $x$.

The causal view is known to face challenges in explaining misrepresentation, where the content of representation does not align with the causal origin (Fodor, 1987). This does not mean that misrepresentation is disconnected from physical causation: misrepresentation can still be causally produced by neural activity or stimuli. What the causal view fails to describe is the content of representation in cases of misrepresentation. For example, consider a subject who sees a rope but mistakenly judges it to be a snake. The natural explanation is that the subject's brain activity or mental content mistakenly represents a snake in the presence of a rope. However, if we adopt the causal view, because the neural state is caused by the rope, the state represents the rope. The causal view therefore fails to characterize this case as a misrepresentation.

## C    TREATMENT OF NOISE

In this section, we discuss the treatment of measurement noise and intrinsic noise in our framework.

We note that properly addressing measurement noise is important in empirical studies. At the same time, our formalization is focused on the simplest case for clarity. Numerical experiments demonstrated that even in the case of artificial NNs, where noiseless measurement is available, extended readout representations were observed. Similar to other feature characterization methods, it could be extended to handle observation noise appropriately.

Our framework has a different view to the previous studies on the so-called intrinsic noise. To highlight the difference, we first describe the typical view on the intrinsic noise, and then describe ours. Typically, studies under causal-view implicitly assume that there is *a single and ideal neural feature* $h = f(x)$ that does not have noise, and that variability across repeated presentations merely reflects mechanistic internal noise, such as probabilistic neuronal spikes. Under this position, the expected value with noise eliminated is regarded as the ideal representation, and the average of multiple observations from repeated presentations of the same stimulus is adopted as the characterization. While we agree that such a mechanistic noise does exist, in contrast, our framework characterizes the representation differently: if multiple activity patterns are observed, we treat all neural features as representations of a signal as long as they allow the recovery of the signal.

## D    EXPERIMENT DETAILS

This section provides the details of our experiments, including the feature inversion (Appendix D.1), feature forwarding (Appendix D.2), feature perturbation (Appendix D.3), computational resources (Appendix D.4), model specifications (Appendix D.5), and dataset (Appendix D.6).

### D.1    FEATURE INVERSION

We used feature inversion as the primary readout procedure $\pi$. Feature inversion uses gradient-based optimization to find an input that produces a feature similar to a given target feature. We describe the problem formulation in Appendix D.1.1 and the optimization methods in Appendix D.1.2.

### D.1.1    PROBLEM FORMULATION

In our feature inversion experiments, we aim to recover the original input from a given neural feature, referred to as the target feature. Feature inversion uses gradient-based optimization to find an input that minimizes the distance between the feature of the input and the target feature. Let $f : X \to H$ denote a neural network that maps an input $x \in X$ to its corresponding feature $h \in H$. Given a target feature $h \in H \simeq \mathbb{R}^d$, the feature inversion is formulated as finding a set of inputs whose features minimize a predefined loss:

$$\pi(h) = \arg\min_{x \in X} \mathcal{L}(f(x), h),$$

where $\mathcal{L}$ denotes a loss function that quantifies the dissimilarity between the feature vector $f(x)$ and the target feature $h$. The solution set $\pi(h)$ contains inputs that yield features close to the target feature.

For the loss function, we used the mean squared error (MSE) in vision models:

$$\mathcal{L}(h_{\text{recon}}, h_{\text{target}}) = \|h_{\text{target}} - h_{\text{recon}}\|_2^2,$$

and the linear combination of MSE and cosine loss in language models:

$$\mathcal{L}(h_{\text{recon}}, h_{\text{target}}) = \|h_{\text{target}} - h_{\text{recon}}\|_2^2 - \cos(h_{\text{target}}, h_{\text{recon}}).$$

### D.1.2 METHODS

To solve the minimization problem defined in Appendix D.1.1, we adopt gradient descent optimization. Instead of directly optimizing the input $x \in X$, we optimize it through iteratively optimizing a latent variable $z \in Z$. Specifically, we define a generator function $g : Z \to X$ that maps the latent variable to the input space. Each step of the optimization process is as follows:

$$z \leftarrow z - \eta \nabla_z \mathcal{L}(f(g(z)), h), \quad x \leftarrow g(z),$$

where $\eta$ is the learning rate.

**Vision models.** We use Deep Image Prior (DIP) (Ulyanov et al., 2020) as the generator $g$. Its parameter $z$ is optimized to produce the input image $x$ from a fixed random inputs. The architectural bias of DIP acts as a structural prior, suppressing high-frequency artifacts and improving perceptual quality. We choose DIP because it imposes minimal prior assumptions about the image distribution and does not require any training data, making it suitable for our analysis. Specifically, we deliberately avoided using pre-trained generative models as a prior, such as GANs and diffusion models, which may introduce biases from their training data and limit the generality of our findings. As an ablation, we also performed feature inversion without DIP and report results in the Appendix G.

**Language models.** For language models, the input is a discrete token sequence $x \in X = \{1, \ldots, V\}^T$, where $V$ denotes the vocabulary size and $T$ the sequence length. For the readout operation, we relax this discrete space to continuous space since the gradient-based optimization in discrete space is not trivial. Instead of directly optimizing the input tokens, we optimize the token logits $z \in \mathbb{R}^{T \times V}$, which are then converted to the continuous tokens $x \in \mathbb{R}^{T \times V}$ as

$$x_i = g(z_i) = \text{softmax}(z_i), \quad \text{for } i = 1, \ldots, T,$$

where the generator function $g$ applies a row-wise softmax to $z$ and each $x_i$ represents the relaxed categorical distribution over the vocabulary at position $i \in [T]$. The final token sequence after gradient-based optimization is obtained by taking the argmax over each row of the optimized logit matrix:

$$x_i = \arg\max_j z_{i,j}, \quad \text{for } i = 1, \ldots, T.$$

In both vision and language modalities, we used the AdamW optimizer from PyTorch. Table 1 summarizes the optimizer settings and training configurations used for feature inversion. Default PyTorch parameters were used for AdamW if not specified. In vision modality, we used a linear learning rate scheduler, which decayed the learning rate to zero over the course of training.

Table 1: Optimizer and training parameters used in feature inversion experiments.

| Parameter | Vision | Language |
|---|---|---|
| Optimization Target | DIP Latent | Token Logits |
| Learning Rate | 0.0001 | 0.1 |
| Iterations | 10,000 | 10,000 |

### D.2 FEATURE FORWARDING

In our supplementary experiment using a variational autoencoder (VAE) model, we additionally adopted another readout method which we call feature forwarding. Given the neural features of the VAE, we directly pass it through the decoder module of the model to reconstruct the image from it. This approach leverages the learned generative capabilities of the VAE, providing an alternative to optimization-based feature inversion. We provide results using this method in Appendix E.

### D.3 FEATURE PERTURBATION

To systematically evaluate the extent of readout representations, we perturbed feature vectors by adding Gaussian noise with calibrated variance. Given a feature vector $h \in \mathbb{R}^d$, we generated a perturbed feature $h' = h + \varepsilon$, where $\varepsilon \sim \mathcal{N}(0, \sigma^2 I_d)$.

The variance $\sigma^2$ was analytically determined to produce a target correlation distance $d_H(h, h') = c \in (0, 1)$ between the original and perturbed features. Under high-dimensional assumptions, the expected correlation distance can be approximated as:

$$c \approx 1 - \frac{1}{\sqrt{1 + \frac{\sigma^2}{\mathrm{Var}(h)}}},$$

where, $\mathrm{Var}(h)$ denotes the sample variance of the feature vector $h$. Solving for $\sigma^2$ gives:

$$\sigma^2 = \mathrm{Var}(h) \left( \frac{1}{(1-c)^2} - 1 \right).$$

This allowed control over perturbation magnitude. We used 10 correlation distance levels: $\{0.1, 0.2, 0.3, 0.4, 0.5, 0.6, 0.7, 0.8, 0.9, 0.99\}$.

### D.4 COMPUTATIONAL RESOURCES

Our experiments flexibly scale with available resources by adjusting key parameters such as the number of noise levels and random seeds, and the number of parallel execution. As a reference, a minimal experiment requires approximately 10 GB of GPU memory and 20 minutes of runtime on an NVIDIA V100 GPU, which reconstructs a single image from one VGG19 layer with 10 noise levels, a number of parallel execution of 10, and one random seed. Full-scale experiments, including all layers, models, and repetitions, were conducted using the following hardware:

- **Local resources:** NVIDIA Tesla V100S (32 GB), Quadro RTX 8000 (48 GB), RTX A6000 (48 GB), and A100 GPUs.

- **Cloud resources:** AWS g5.48xlarge instances equipped with 8 NVIDIA A10G Tensor Core GPUs, ABCI (AI Bridging Cloud Infrastructure) rt_HF nodes equipped with 8 NVIDIA H200 SXM 141GB GPUs.

### D.5 MODEL SPECIFICATIONS

We evaluated representations of deep neural network models across a range of pretrained vision and language models. For the vision modality, we used VGG19, CLIP (Base Patch32, Large Patch14), DINOv2 (Base, Large, Giant), and SDXL-VAE. For the language modality, we used BERT (Base), GPT2 (Small, Medium, Large, XL), and OPT (125M, 350M, 1.3B, 2.7B). For each model, we selected representative layers for analysis: all 16 convolutional layers in VGG19, and one layer per quarter depth in other models (e.g., layers 0, 3, 6, 9 in a 12-layer model). Except for VGG19, all models weights were obtained from the HuggingFace Transformers library, and we provide details on the source of each model below.

**VGG19** We used the weights provided at the following URL: https://figshare.com/ndownloader/files/38225868. This weight is a PyTorch-compatible conversion of the one originally provided in Caffe (http://www.robots.ox.ac.uk/~vgg/software/very_deep/caffe/VGG_ILSVRC_19_layers.caffemodel).

**Other models** Table 2 lists HuggingFace model identifiers for each model.

Table 2: HuggingFace model identifiers for the transformer models used in our experiments.

| Model Family | HuggingFace Identifier |
|---|---|
| DINOv2 (ViT) | facebook/dinov2-base |
| | facebook/dinov2-large |
| | facebook/dinov2-giant |
| CLIP (ViT) | openai/clip-vit-base-patch32 |
| | openai/clip-vit-large-patch14 |
| SDXL-VAE (VAE) | stabilityai/sdxl-vae |
| BERT | google-bert/bert-base-uncased |
| GPT2 | openai-community/gpt2 |
| | openai-community/gpt2-medium |
| | openai-community/gpt2-large |
| | openai-community/gpt2-xl |
| OPT | facebook/opt-125m |
| | facebook/opt-350m |
| | facebook/opt-1.3b |
| | facebook/opt-2.7b |

## D.6 DATASET

For the vision modality, we randomly sampled 64 natural images for our primary experiments (Figure 7) from the test-split of the ImageNet dataset via HuggingFace datasets (ILSVRC/imagenet-1k).

To analyze differences in representation size between correctly classified images ("hit") (Figure 8) and incorrectly classified ones ("miss") by VGG19 (Figure 9), we sampled 16 images from the ImageNet validation set in total. Hit images included 8 images where the model's top-1 prediction matched the ground-truth label. Miss images included 8 images where the true label was not within the top-5 predictions. We excluded ambiguous cases to ensure label clarity.

For the experiment with noise images, we generated noise images by sampling pixel values uniformly from [0, 255] (Figure 10). We prepared four images with different random seeds for robustness.

For the language modality, we sampled 64 sequences from the validation split of the C4 dataset via HuggingFace Datasets (allenai/c4). Each sequence was truncated to a maximum of 256 tokens in order to control computational costs.

In all experiments, the dataset was curated beforehand, and no post-hoc cherry-picking was performed.

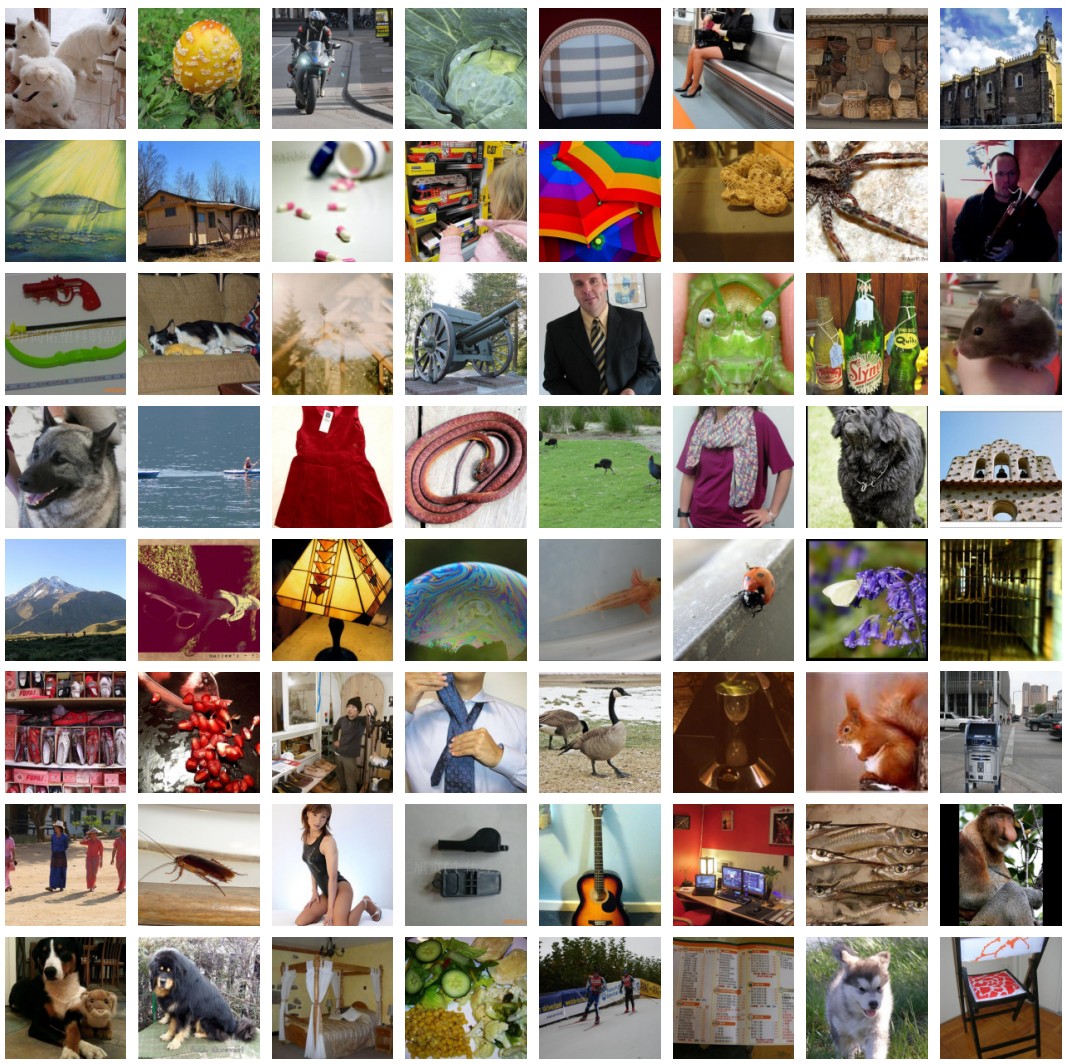

Figure 7: 64 natural images used in the vision experiments, randomly sampled from the test-split of ImageNet dataset. The images were selected prior to analysis and used consistently across experiments without post-hoc cherry-picking.

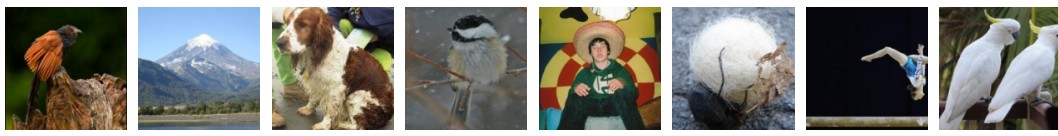

Figure 8: Images that were correctly classified by VGG19 on the ImageNet validation set ("hit"). Each image was correctly classified based on the top-1 prediction of VGG19.

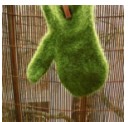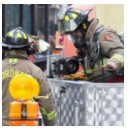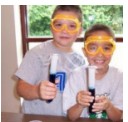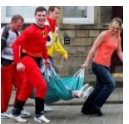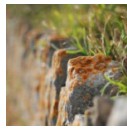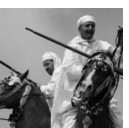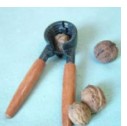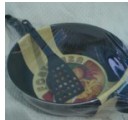

Figure 9: Images that were incorrectly classified by VGG19 on the ImageNet validation set ("miss"). Each image was misclassified as the correct label was not included in the top-5 predictions of VGG19.

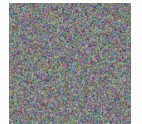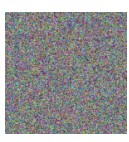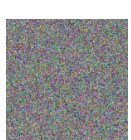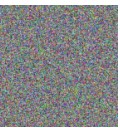

Figure 10: Random noise images used for representation size analysis. Each image was generated by independently sampling pixel values from a uniform distribution over [0, 255] with different random seeds.

# E    DETAILS OF RESULTS FOR VISION MODELS

We presents additional results for the vision modality to complement the main text. We provide reconstructed images, and quantitative evaluations.

## E.1    RECONSTRUCTED IMAGES

This section presents reconstructed images from perturbed features for each vision model.

### E.1.1    VGG19

We provide results of four representative images out of 64 samples (Figure 11 to Figure 14) for space constraints.

Feature distance $d_H$

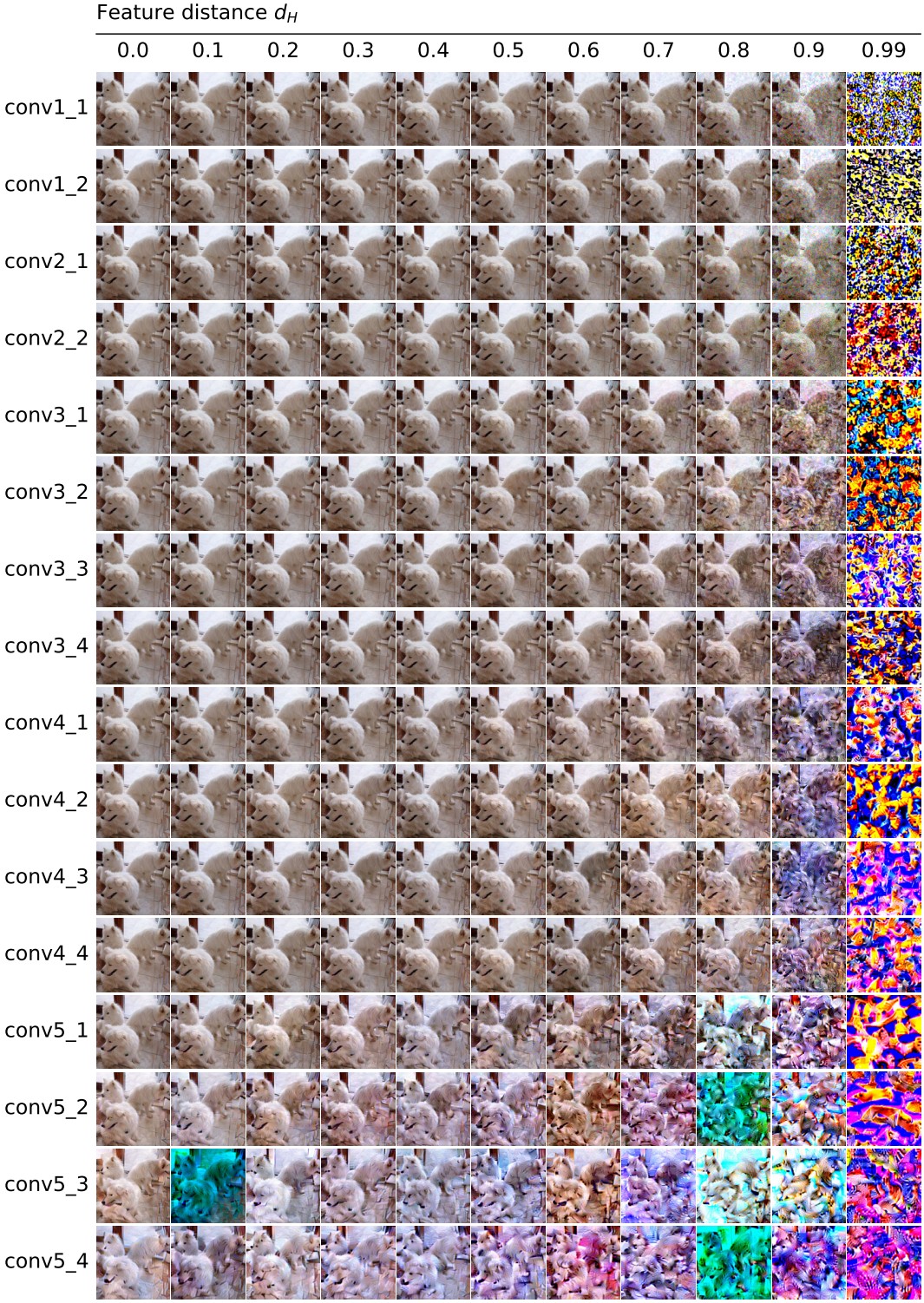

Figure 11: Images reconstructed from perturbed VGG19 features. Each row corresponds to a convolutional layer and each column corresponds to a magnitude of feature perturbation, measured by the correlation distance $d_H$ between the original and perturbed features. Even under substantial perturbations, the reconstructed images remain faithful in early to middle layers.

Feature distance $d_H$

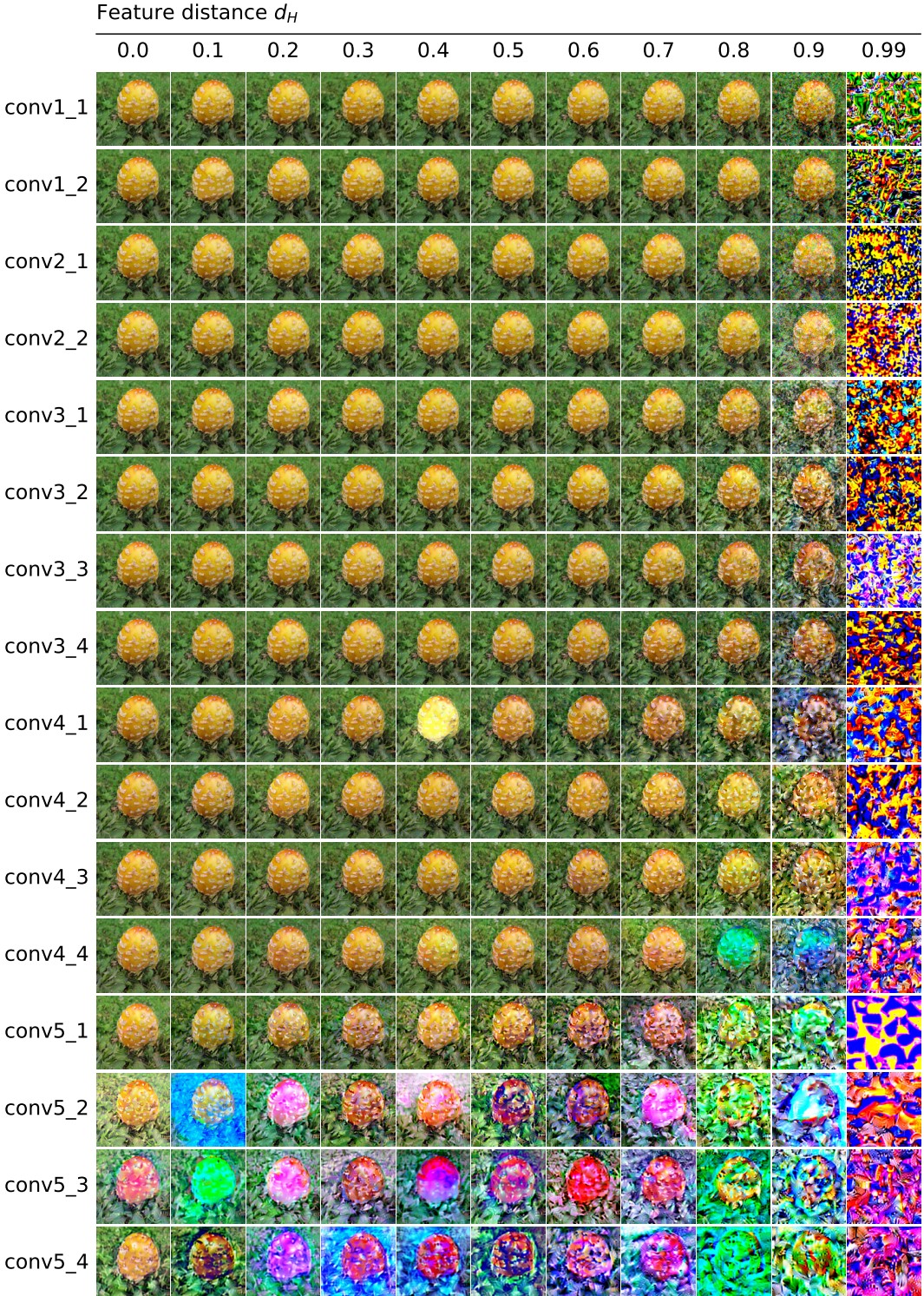

Figure 12: Images reconstructed from perturbed VGG19 features. Layout follows Figure 11.

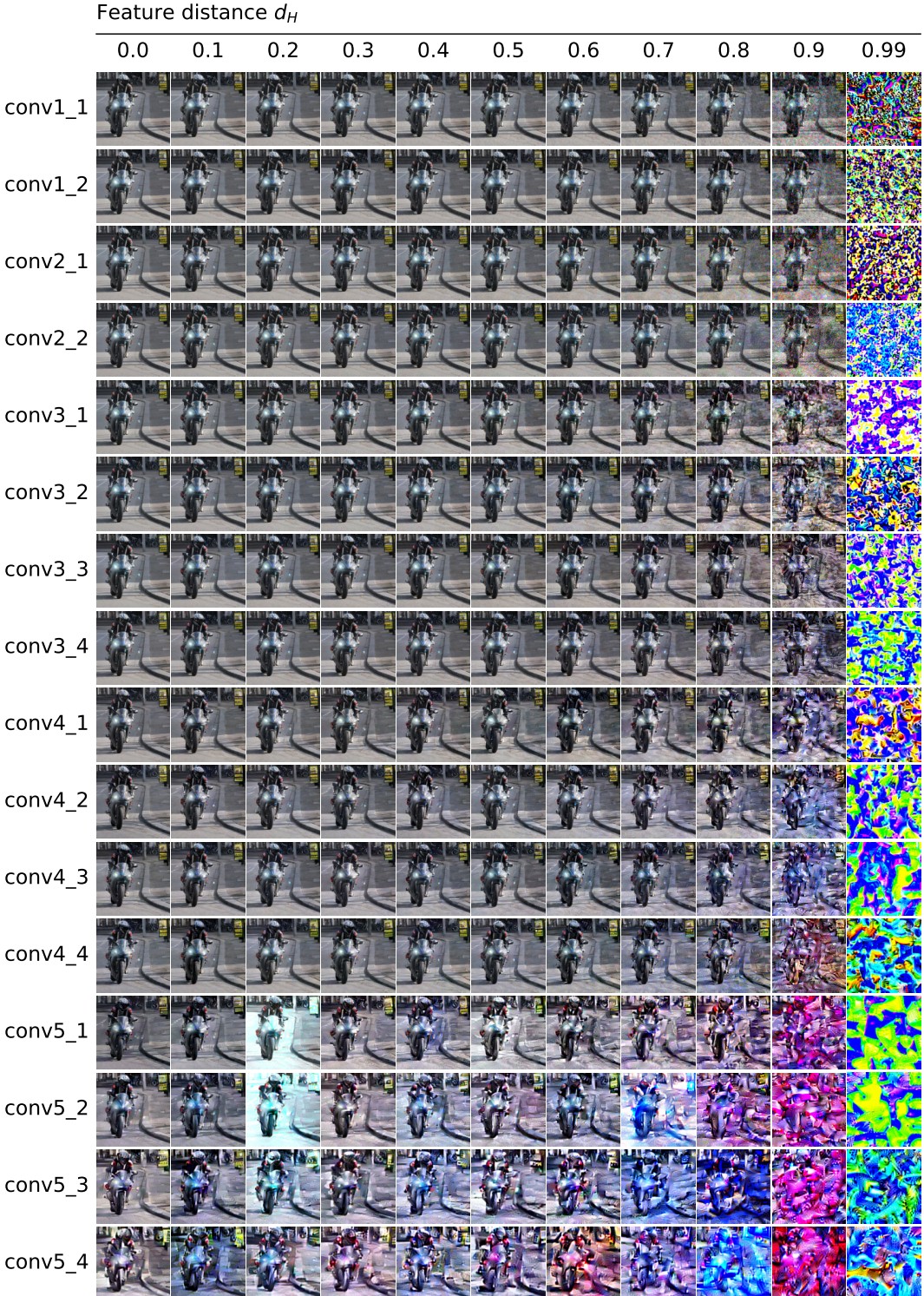

Figure 13: Images reconstructed from perturbed VGG19 features. Layout follows Figure 11.

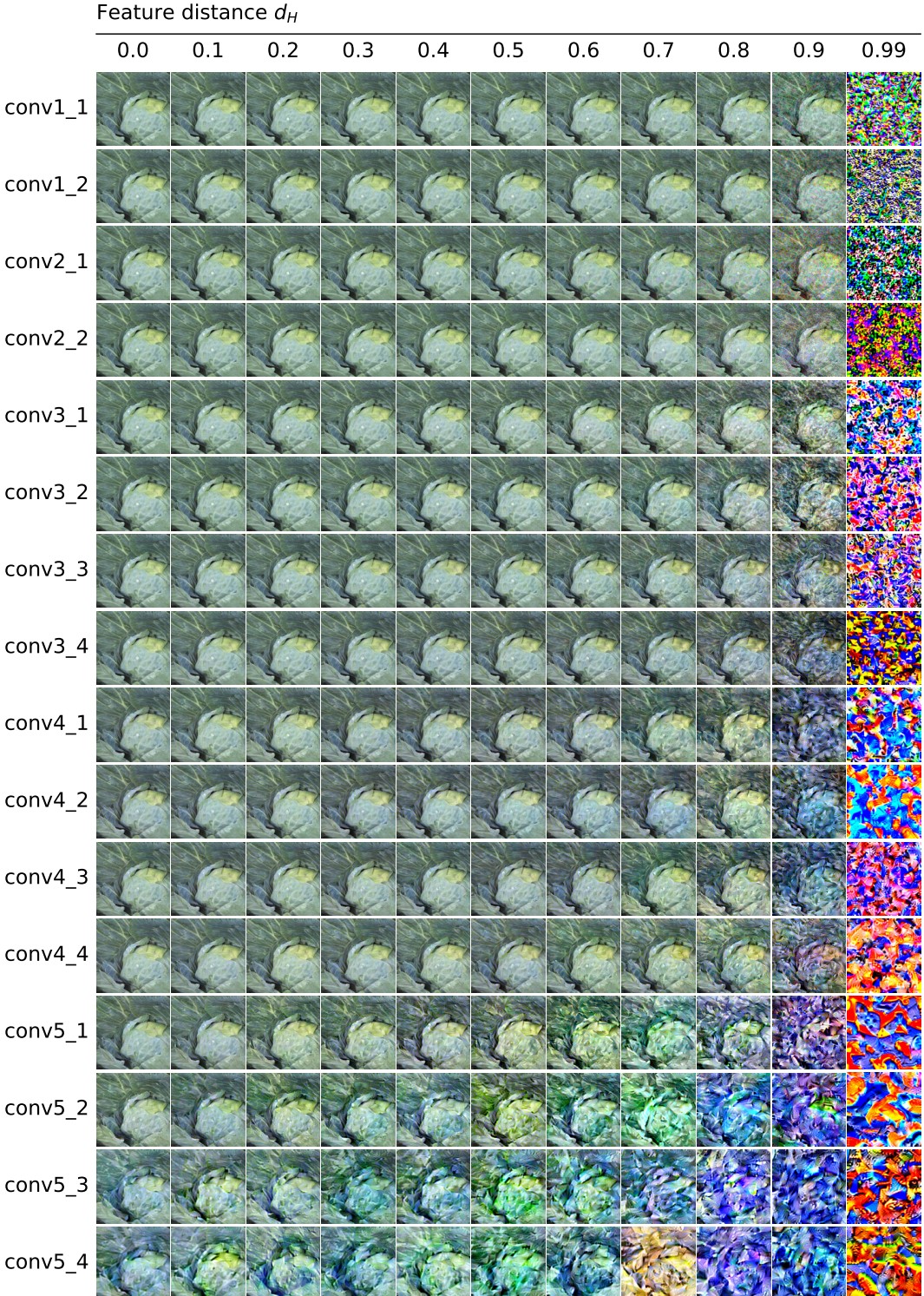

Figure 14: Images reconstructed from perturbed VGG19 features. Layout follows Figure 11.

### E.1.2 DINOv2

We provide images reconstructed from perturbed features of DINOv2-giant in Figure 15. We provide reconstructions of one sample due to space constraints.

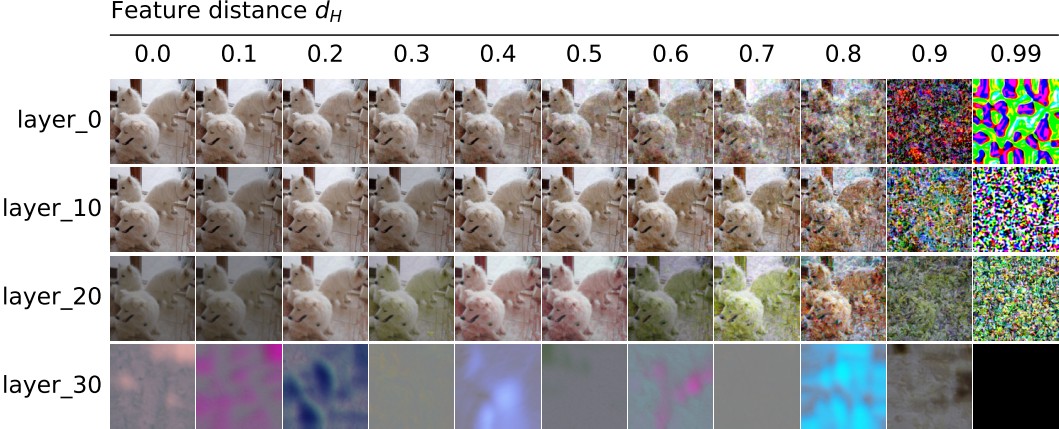

Figure 15: Images reconstructed from perturbed DINOv2-giant features. Layout follows Figure 11.

### E.1.3 CLIP

We provide images reconstructed from perturbed features of CLIP ViT-L/14 in Figure 16. We provide reconstructions of one sample image due to space constraints.

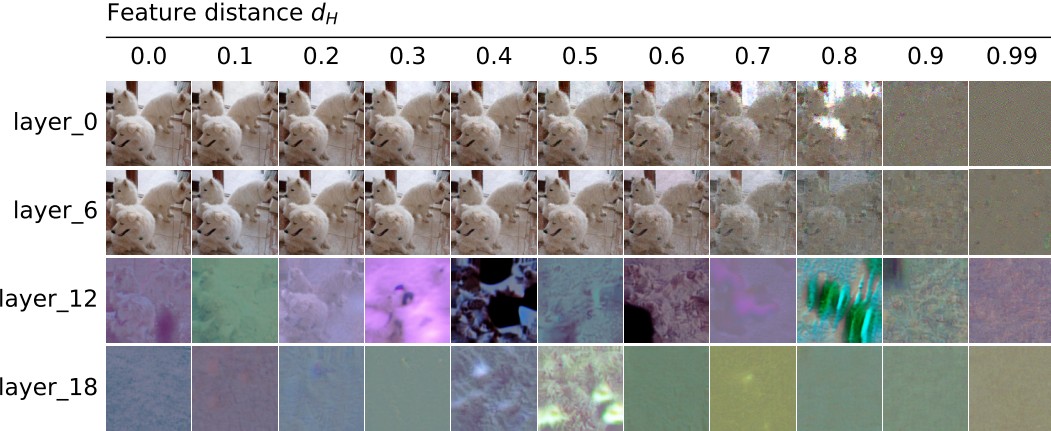

Figure 16: Images reconstructed from perturbed CLIP ViT-L/14 features. Layout follows Figure 11.

### E.1.4 SDXL-VAE

We provide images reconstructed from perturbed features of SDXL-VAE in Figure 17 and Figure 18 for feature inversion and feature forwarding methods, respectively. We provide reconstructions of four sample image due to space constraints.

Feature distance $d_H$

| 0.0 | 0.1 | 0.2 | 0.3 | 0.4 | 0.5 | 0.6 | 0.7 | 0.8 | 0.9 | 0.99 |
| --- | --- | --- | --- | --- | --- | --- | --- | --- | --- | --- |

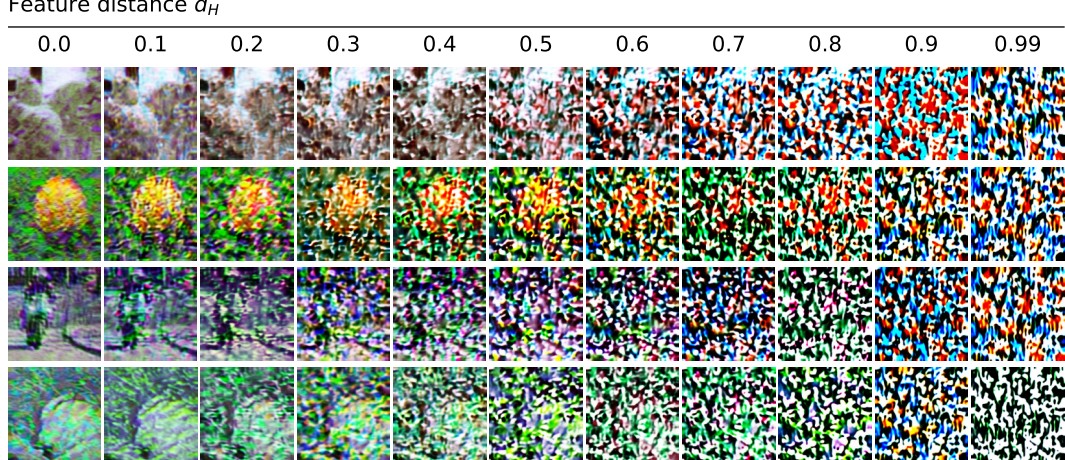

Figure 17: Images reconstructed from perturbed SDXL-VAE features using feature inversion. Each row corresponds to an image and each column shows reconstructions from features perturbed at increasing correlation distances $d_H \in \{0.0, 0.1, \ldots, 0.9, 0.99\}$.

Feature distance $d_H$

| 0.0 | 0.1 | 0.2 | 0.3 | 0.4 | 0.5 | 0.6 | 0.7 | 0.8 | 0.9 | 0.99 |
| --- | --- | --- | --- | --- | --- | --- | --- | --- | --- | --- |

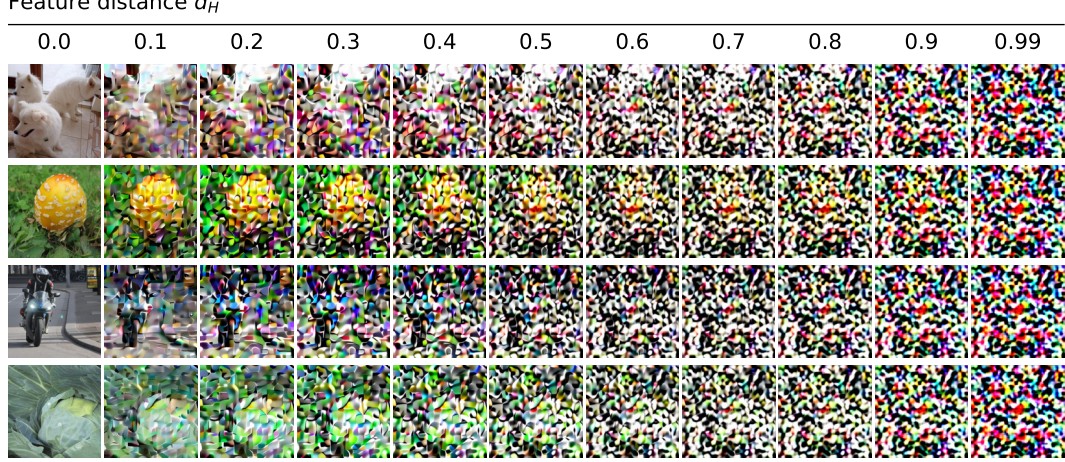

Figure 18: Images reconstructed from perturbed SDXL-VAE features using feature forwarding. Figure layout follows Figure 17.

## E.2 QUANTITATIVE RESULTS

This section provides additional quantitative results for reconstruction experiments.

### E.2.1 VGG19

We present quantitative results of all 64 images for all 16 convolutional layers of VGG19 in Figure 19. We also provide results of perceptual metrics (SSIM, PSNR, LPIPS, DISTS) in Figure 20.

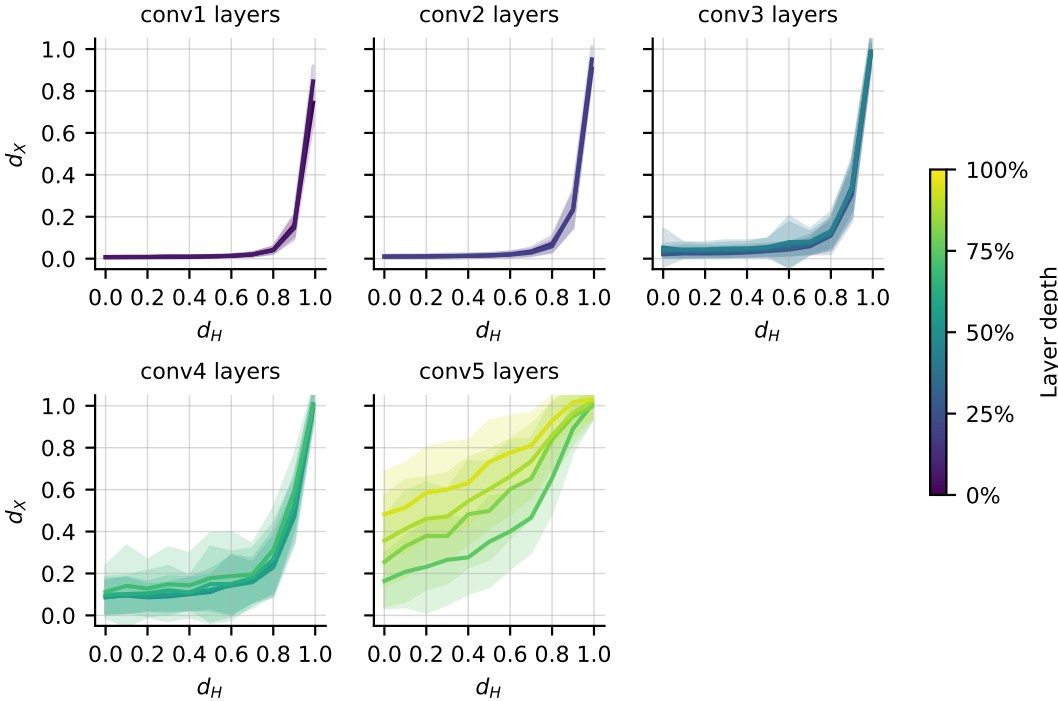

Figure 19: Quantitative results of VGG19. Pixel correlation distance ($d_X$) between reconstructed and original images plotted against feature correlation distance ($d_H$) between the perturbed and original features. Each subplot corresponds to a group of convolutional layers, and line colors indicate layer depth within the group.

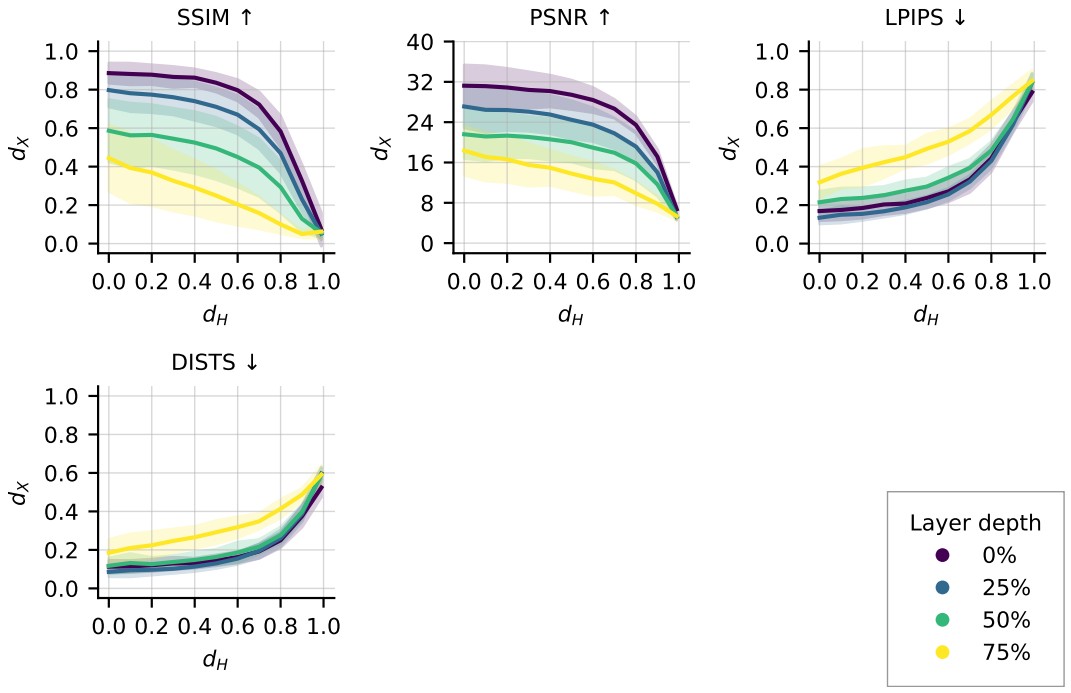

Figure 20: Results of perceptual metrics (SSIM, PSNR, LPIPS, DISTS) of VGG19. Perceptual metrics between reconstructed and original images plotted against feature correlation distance ($d_H$) between the perturbed and original features. Each subplot corresponds to a metric. Quarter-depth layers are shown with colors indicating layer depth.

### E.2.2 DINOv2

We present quantitative results of all 64 natural images for all three model sizes we tested (base, large, giant) in Figure 21.

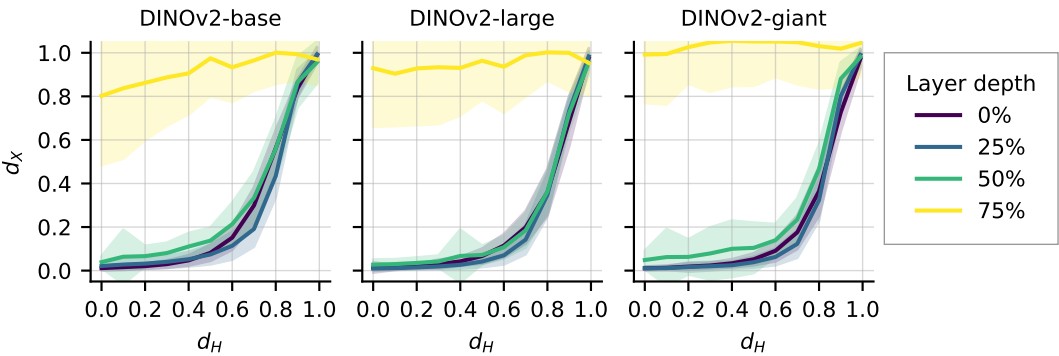

Figure 21: Results of DINOv2 models. Pixel correlation distance ($d_X$) between reconstructed and original images plotted against feature correlation distance ($d_H$) between the perturbed and original features. Each subplot corresponds to a model size. Quarter-depth layers are shown with colors indicating layer depth.

### E.2.3 CLIP

We present quantitative results of all 64 natural images for all two model sizes we tested (base, large) in Figure 22

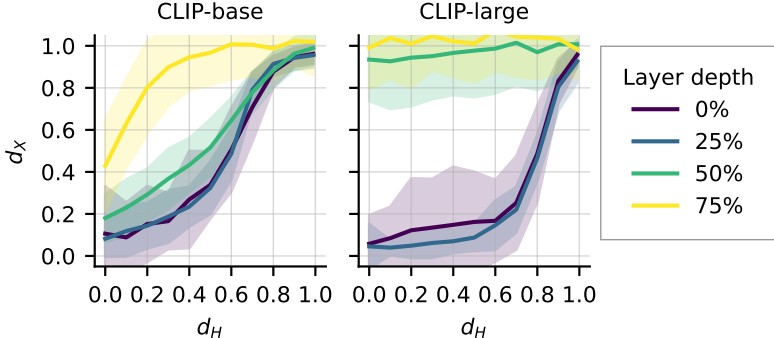

Figure 22: Results of CLIP models. Pixel correlation distance ($d_X$) between reconstructed and original images plotted against feature correlation distance ($d_H$) between the perturbed and original features. Each subplot corresponds to a model size. Quarter-depth layers are shown with colors indicating layer depth.

### E.2.4 SDXL-VAE

We present quantitative results of all 64 natural images for SDXL-VAE using two methods, feature inversion and feature forwarding, in Figure 23.

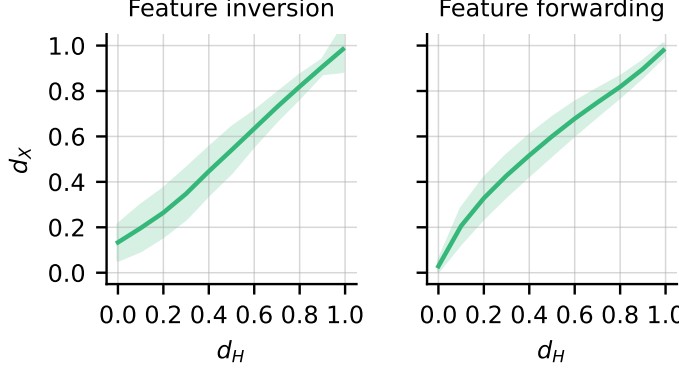

Figure 23: Results of SDXL-VAE with feature inversion and feature forwarding. Pixel correlation distance ($d_X$) between reconstructed and original images plotted against feature correlation distance ($d_H$) between the perturbed and original features. Each subplot corresponds to a readout method.

## F ADDITIONAL RESULTS FOR APPLICATION OF REPRESENTATION SIZE

In order to examine if the observed differences in representation sizes (Section 4.2) are due to the effect of training, we performed an additional analysis. Specifically, we repeated the comparison of representation sizes of the same set of images using VGG19 with randomly initialized weights, which we refer to as the untrained VGG19.

As a result, we observed the similar differences in representation sizes in the untrained model (Figure 24A). In the untrained VGG19, hit images have shown larger representation sizes than miss images, and noise images have shown zero representation sizes across layers. This similarity across trained and untrained models suggests that the observed differences in representation sizes reflect architectural biases of the model, rather than solely training effects. Also, the comparison between trained and untrained VGG19 using natural images showed that the trained model exhibits larger representation sizes, particularly in middle to higher layers (Figure 24B). This demonstrates that training on natural images increases the representation sizes of the corresponding image type. In summary, we found that architectual biases of the model underlie the observed differences in representation sizes, and the training expands the representation sizes for the trained image type.

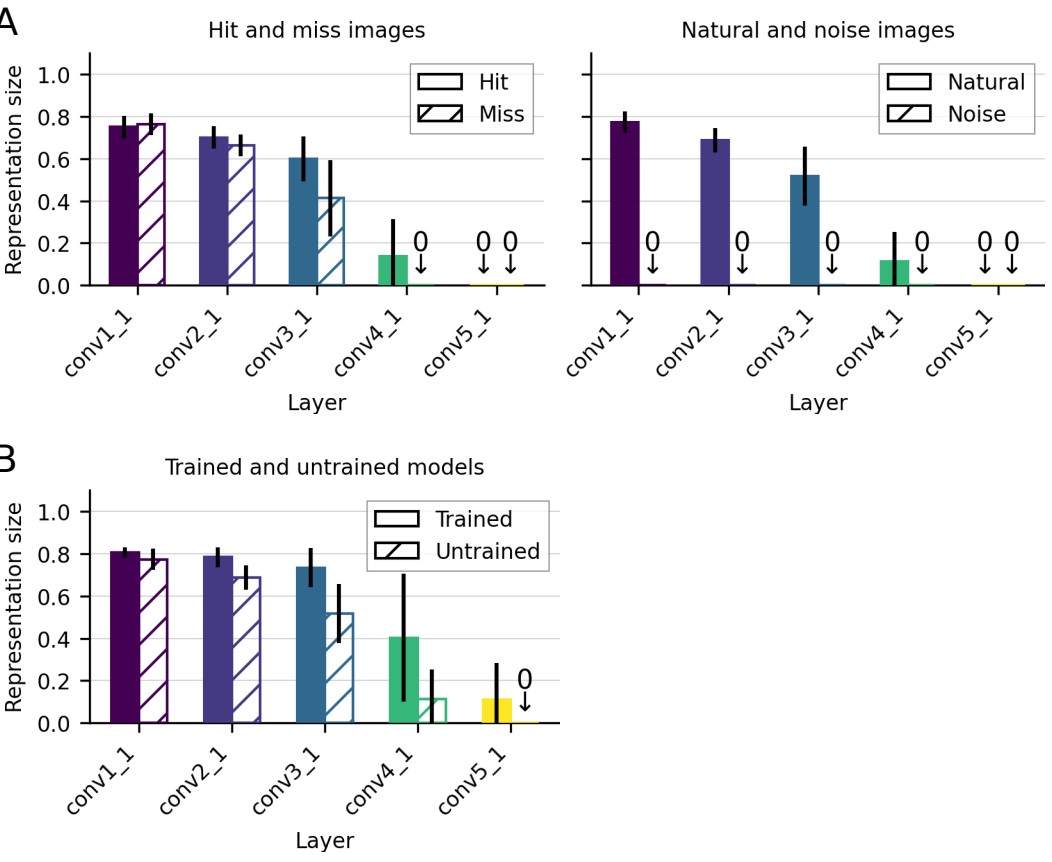

Figure 24: Comparison of representation sizes of different images types using an untrained VGG19 model. Bars show the mean representation sizes across samples, with error bars indicating $\pm 1$ SE. **A**: The hit images show larger representation sizes on untrained VGG19 as well, indicating these differences reflect architectural biases, not solely training effects. **B**: Comparison between trained and untrained VGG19 using natural images showed that the trained model exhibits larger sizes, particularly in middle to higher layers, demonstrating the effect of training on representation sizes.

## G    ABLATION STUDIES ON THE EFFECT OF DIP

To exclude the possibility that our main results are solely due to the denoising capability of DIP, we performed two ablation studies to assess the effect of using Deep Image Prior (DIP) as the generator in feature inversion. First, we ablated DIP entirely by removing the generator and directly optimizing pixel values, and report results in the Appendix G.1. Second, we examined the effect of DIP to denoise perturbed images by ablating the neural network in the feature inversion experiment, considering the pixel space itself as the feature space, and report results in the Appendix G.2. Both experiments indicates that DIP alone cannot account for the high representation sizes observed in our main results.

### G.1    RECONSTRUCTION WITHOUT DIP

We consider applying feature inversion without a generator $g$. Specifically, we initialize the pixel values of a reconstructed image $x$ with random values, and optimize the pixel values using gradient:

$$x \leftarrow x - \eta \nabla_x \mathcal{L}(f(x), h). \tag{1}$$

For optimization parameters, we set the learning rate to 0.01. Other hyperparameters followed those in Table 1.

We present reconstructed images using pixel optimization in Figure 25 to Figure 28. We present 4 samples out of 64 samples due to space constraints. Even without DIP, the reconstructed images retains perceptual similarity to the original images under significant perturbations in early to middle layers. Specifically, these images retain the fine-grained information suffices to perform human recognition. We present quantitative results of all 64 images for all 16 convolutional layers of VGG19 in Figure 29. We observed relatively steep increase of pixel correlation distance $d_X$ against feature correlation distance $d_H$ in small perturbation. However, these small perturbations reflects high-frequency variations rather than the loss of essential information for downstream tasks. These results indicate that the recovery of input images from perturbed features are not solely due to the denoising capability of DIP.

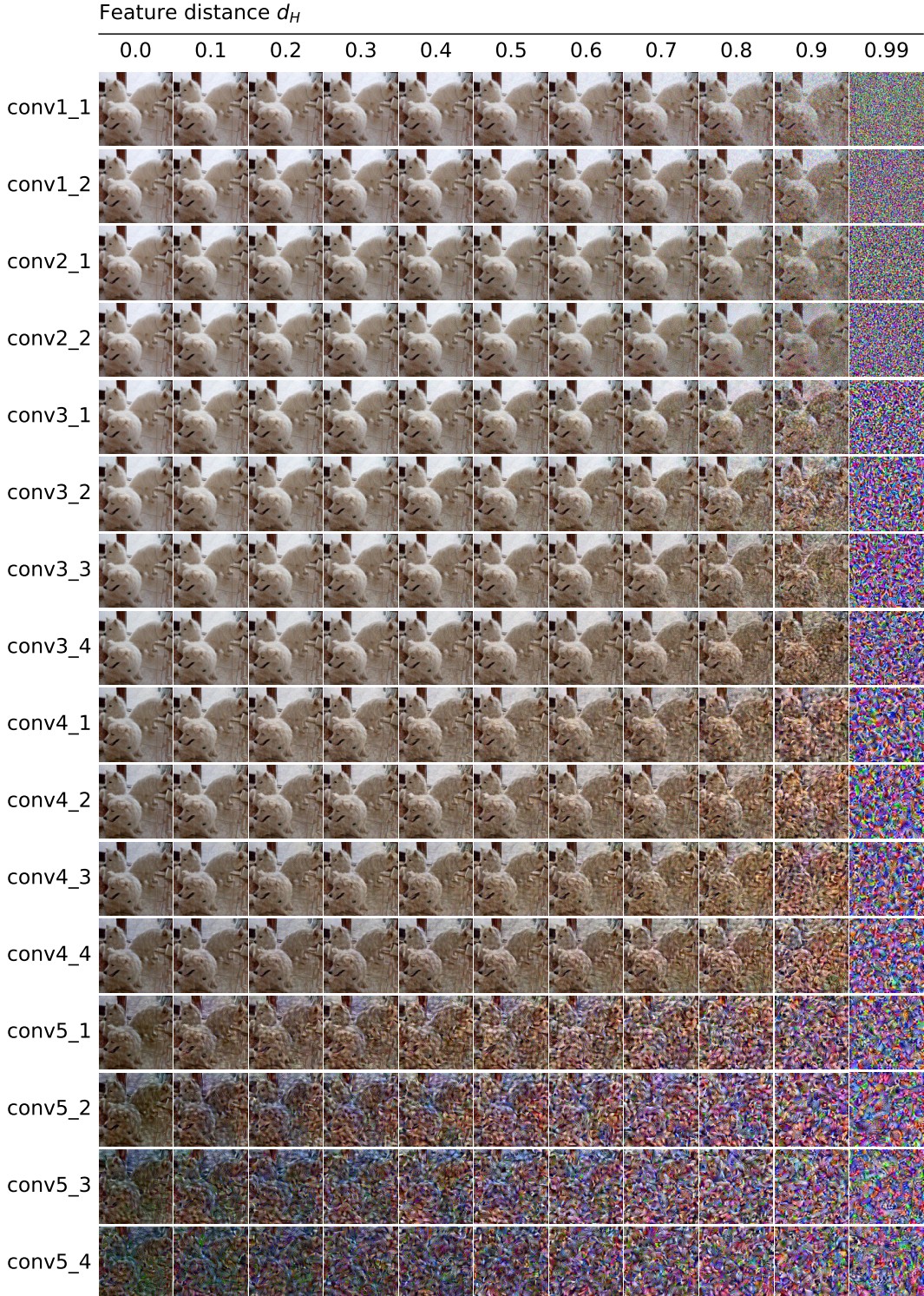

Figure 25: Images reconstructed from perturbed features in VGG19 using pixel optimization, ablating DIP. Layout follows Figure 11.

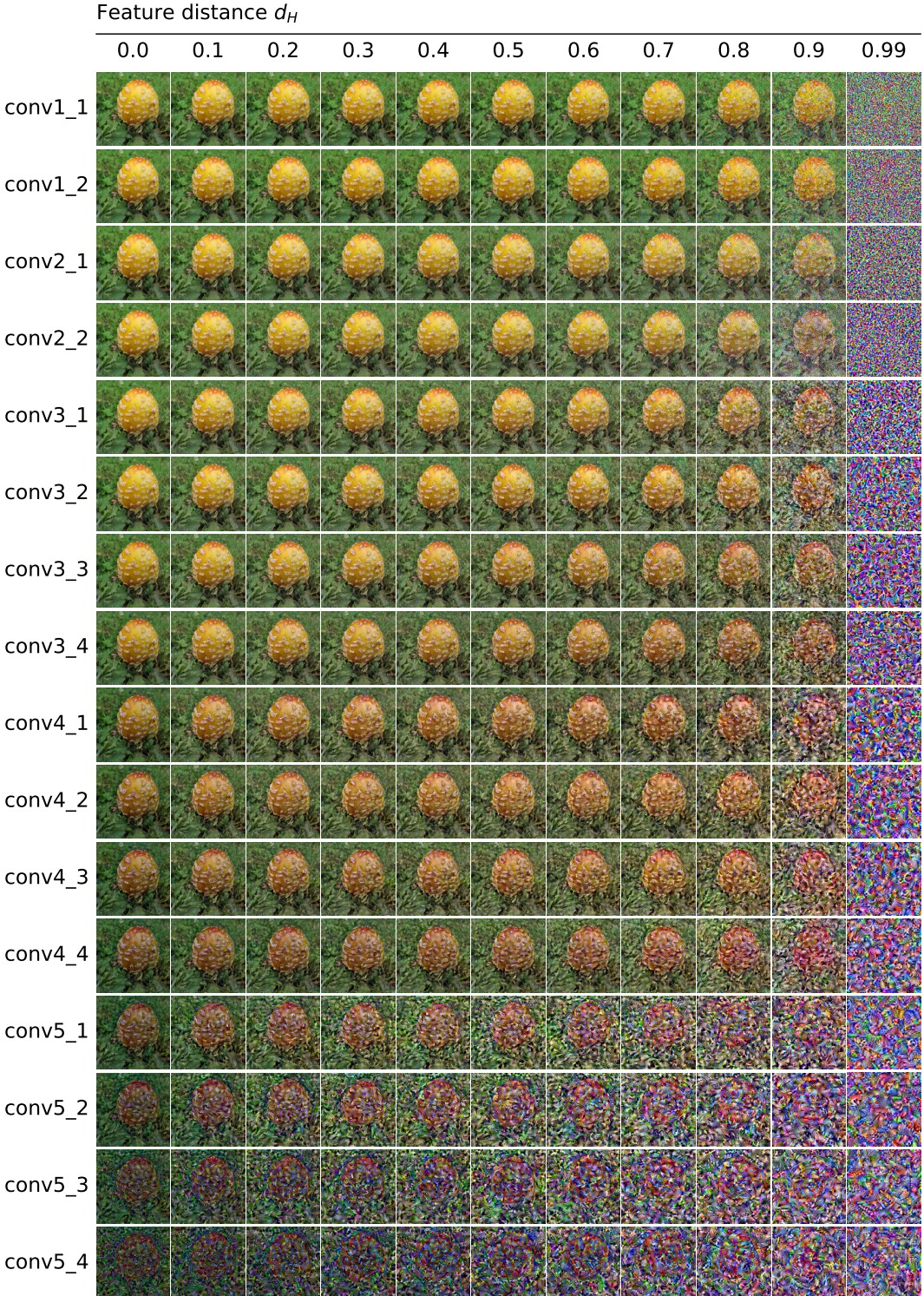

Figure 26: Images reconstructed from perturbed features in VGG19 using pixel optimization, ablating DIP. Layout follows Figure 11.

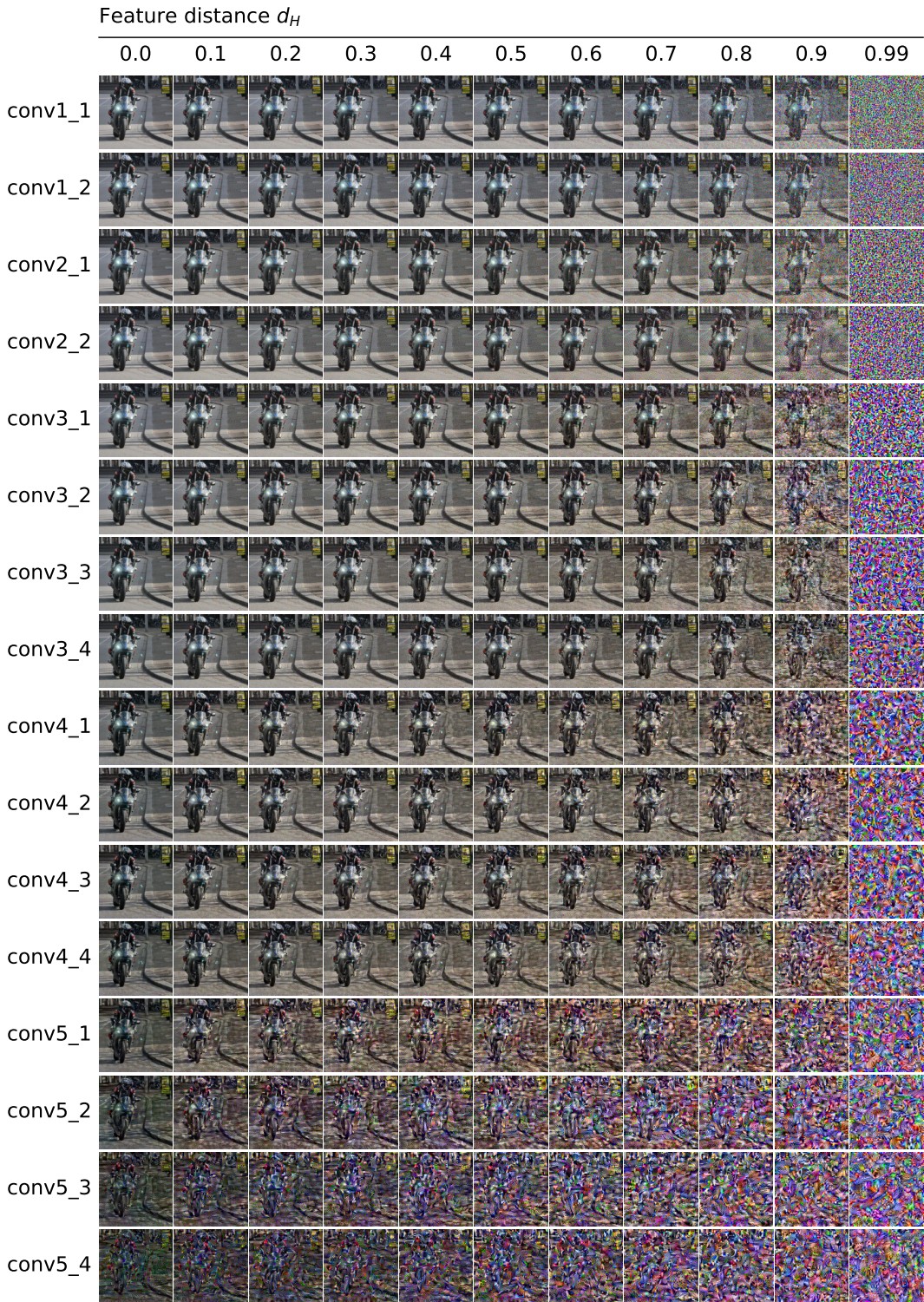

Figure 27: Images reconstructed from perturbed features in VGG19 using pixel optimization, ablating DIP. Layout follows Figure 11.

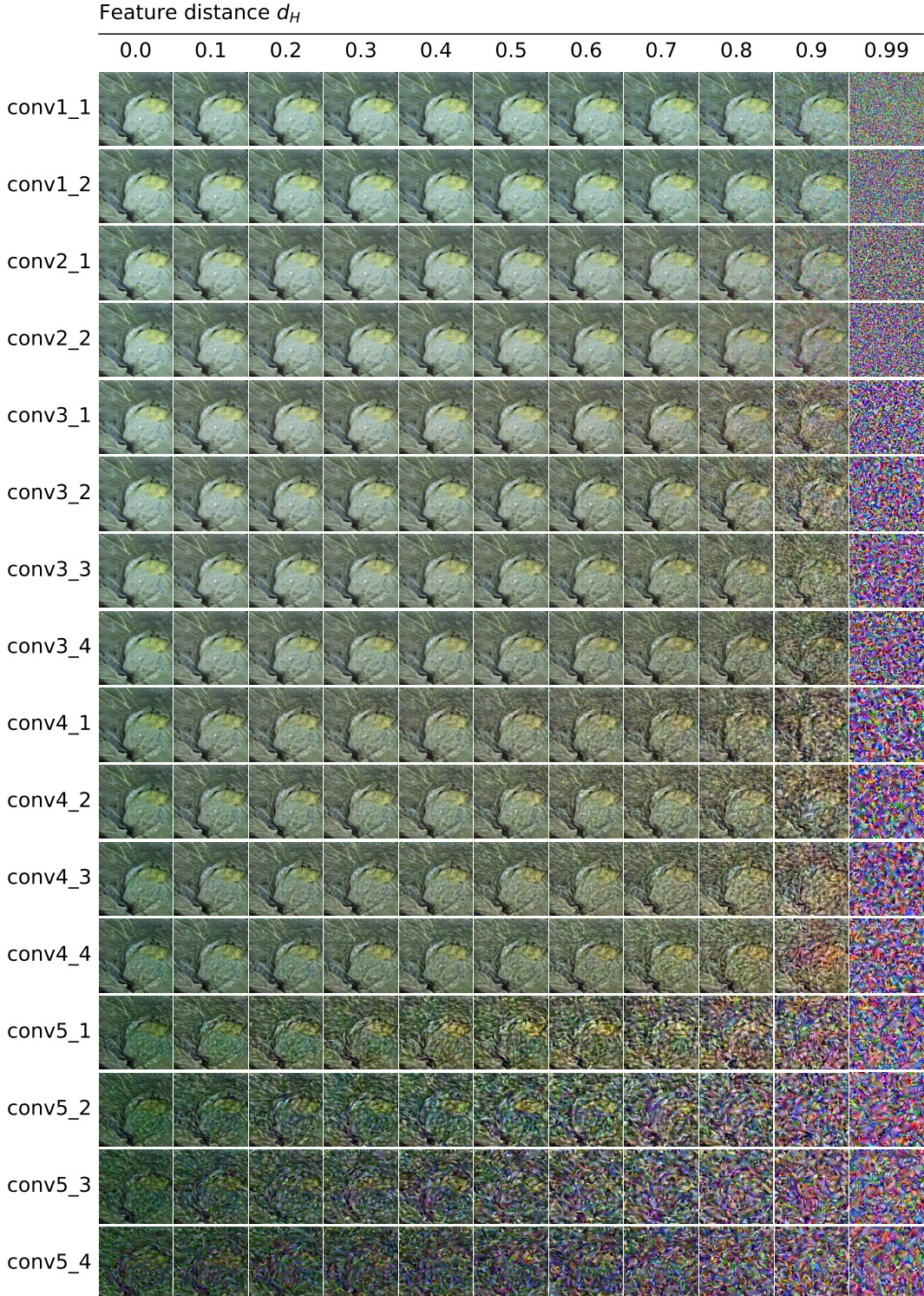

Figure 28: Images reconstructed from perturbed features in VGG19 using pixel optimization, ablating DIP. Layout follows Figure 11.

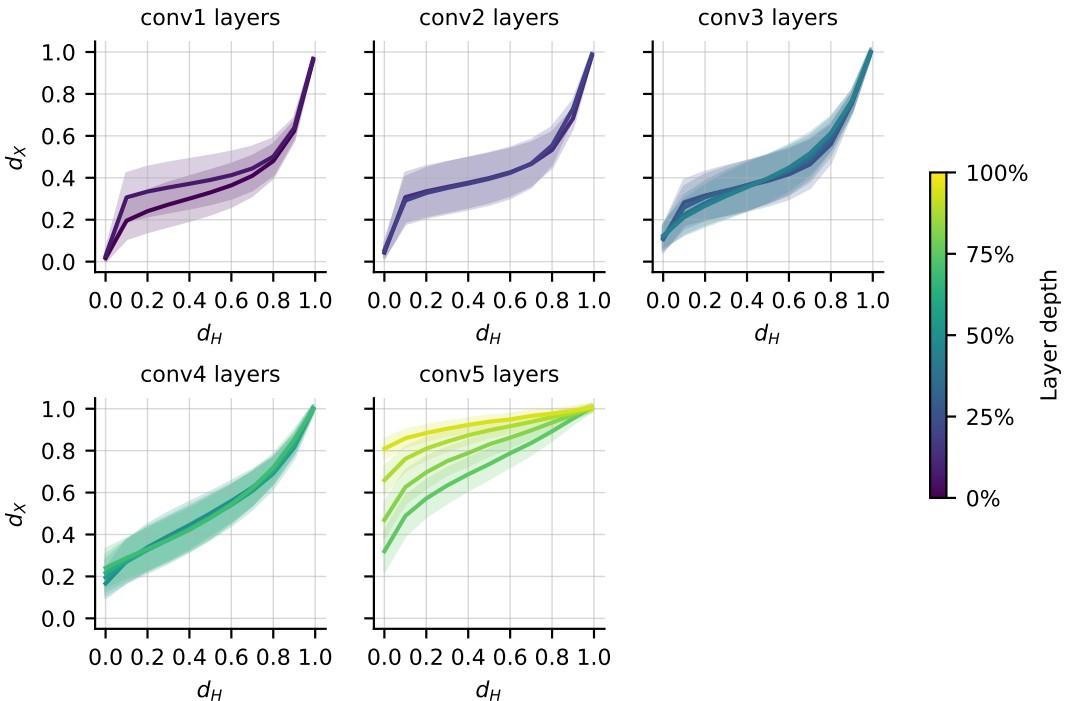

Figure 29: Quantitative results of VGG19 with pixel optimization. Formatting follows Figure 19.

### G.2 REPRESENTATION SIZE ON PIXEL SPACE

We have performed the ablation experiment of measuring the representation size in the input space with DIP. We perturb an image, and then optimize the latent variable of DIP, so that the produced image matches the perturbed one. This is identical to the feature inversion experiment except that the feature space coincides with the pixel space, i.e., ablating the neural network. Optimization parameters are the same as those in Table 1. If our main results presented in the main text and Appendix E were solely due to the denoising capability of DIP, we would expect to see similar representation sizes in this ablation experiment.

Figure 30 and Table 3 present the results of this ablation experiment. While DIP could remove some noise from the perturbed images (Figure 30, left), the results of this ablation (Figure 30, left) are much closer to a straight line than the results of VGG19 (Figure 30, right). This demonstrates that DIP alone cannot account for the large readout representations we observe. In Table 3, we present the result of representation size in comparison to that of the VGG19 layers. The pixel space has representation size of $0.32 \pm 0.047$, which is significantly smaller than the lower and middle layers of VGG19. This gap confirms that DIP alone cannot explain the high representation size observed in those layers.

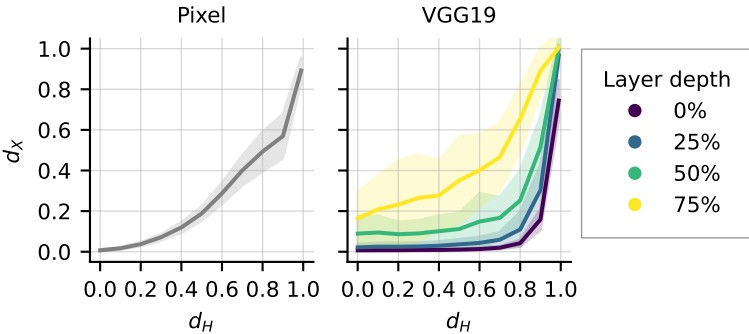

Figure 30: Quantitative results of ablation experiment measuring representation size in the pixel space using DIP. **Left**: Results of the ablation study, representation size in the pixel space using DIP. **Right**: Results of the original experiment using VGG19 and DIP for comparison. Although the DIP could remove some noise from the perturbed images (left), the line is significantly steeper than those of lower and middle layers of VGG19 (right), indicating that DIP alone cannot explain the high representation size observed in those layers.

| Model | Feature space | Size |
|-------|--------------|------|
| – | **Pixel** | $0.32 \pm 0.047$ |
| VGG19 | conv1_1 | $0.81 \pm 0.024$ |
| VGG19 | conv2_1 | $0.78 \pm 0.046$ |
| VGG19 | conv3_1 | $0.73 \pm 0.093$ |
| VGG19 | conv4_1 | $0.40 \pm 0.30$ |
| VGG19 | conv5_1 | $0.11 \pm 0.17$ |

Table 3: Representation size measured in the pixel space compared to those of VGG19 layers. The pixel space has a significantly smaller representation size (top row) than the lower and middle layers of VGG19, indicating that DIP alone cannot explain the high representation size observed in those layers.

## H    DETAILS OF RESULTS FOR LANGUAGE MODELS

We present additional results for the language modality to complement the main text. We provide reconstructed sentences, quantitative evaluations, and insights into the differences in results across models.

### H.1    RECONSTRUCTED SENTENCES

We provide sample of sentences reconstructed from the perturbed features in Table 4 to Table 7. We present results of a single sample sentence using BERT for brevity.

Table 4: Reconstructed text from 0-th layer of google-bert/bert-base-uncased. The top row shows the original text for reference. Subsequent rows present texts reconstructed from perturbed features. The column $d_H$ indicates the correlation distance between the original and perturbed features, while $d_X$ reports the token error rate between the original and reconstructed text. Text is truncated at 500 characters due to space constraints. Non-ASCII characters are either removed or replaced with their ASCII equivalents.

| Type | $d_H$ | $d_X$ | Text |
|------|-------|-------|------|
| Original | – | – | New Zealand welcomes the World Rookie Tour for the second year! Get ready rookies! The 2018/ 2019 World Rookie Tour season has officially begun with the New Zealand Rookie Fest from the 14th to the 16th of August! Following last years success, the Black Yeti returns to New Zealand for the second consecutive year! From Tuesday 14 to Thursday 16 August the amazing Cardrona Alpine Resort in Wanaka will host some of the worlds best under eighteen snowboarders and give them the opportunity to show th |
| Recon | 0.00 | 0.00 | new zealand welcomes the world rookie tour for the second year! get ready rookies! the 2018 / 2019 world rookie tour season has officially begun with the new zealand rookie fest from the 14th to the 16th of august! following last year s success, the black yeti returns to new zealand for the second consecutive year! from tuesday 14 to thursday 16 august the amazing cardrona alpine resort in wanaka will host some of the world s best under eighteen snowboarders and give them the opportunity to sh |
| | 0.20 | 0.00 | new zealand welcomes the world rookie tour for the second year! get ready rookies! the 2018 / 2019 world rookie tour season has officially begun with the new zealand rookie fest from the 14th to the 16th of august! following last year s success, the black yeti returns to new zealand for the second consecutive year! from tuesday 14 to thursday 16 august the amazing cardrona alpine resort in wanaka will host some of the world s best under eighteen snowboarders and give them the opportunity to sh |
| | 0.40 | 0.00 | latest new zealand welcomes the world rookie tour for the second year! get ready rookies! the 2018 / 2019 world rookie tour season has officially begun with the new zealand rookie fest from the 14th to the 16th of august! following last year s success, the black yeti returns to new zealand for the second consecutive year! from tuesday 14 to thursday 16 august the amazing cardrona alpine resort in wanaka will host some of the world s best under eighteen snowboarders and give them the opportunity |
| | 0.80 | 0.57 | ##. inline zealand welcome bodies dove worldloid tour format chestnut second yearryaaj feather rookie genetically contributor disclosed preservation harvest 2019 rover rookie rear season has officially boyd defaultuth brightest zealand indo fest austin gateway bill touring 16thhend kid ul followingada year joe success 21stettes mandir counties ashes returnsusshulncia juno avail second consecutive year! sh tuesdayscreen justification thursday 11 homestead rodney amazing cardrona integration re |
| | 0.99 | 1.00 | ##. womanquentoin sacrifice fry shadowtya accord jonny saloonfinger billhr royal{ ldtori globe analogouslase vanity lex shamthermal edmund differencesnington needlemable refereesbbaistic replacementsosi coats royalssar smokinglett outfit nippon rap tialfbin sacramento register inherited step arabiangenic collectionvidlinger wileycorp flipbution rosewood yan incorporated formats pei giblelatingjal other promised proceeds12ticamah expandsahricuttererry / revueffa kato commencementdies advised |

Table 5: Reconstructed text from 3rd layer of google-bert/bert-base-uncased. Formatting follows Table 4.

| Type | $d_H$ | $d_X$ | Text |
|---|---|---|---|
| Original | – | – | New Zealand welcomes the World Rookie Tour for the second year! Get ready rookies! The 2018/ 2019 World Rookie Tour season has officially begun with the New Zealand Rookie Fest from the 14th to the 16th of August! Following last years success, the Black Yeti returns to New Zealand for the second consecutive year! From Tuesday 14 to Thursday 16 August the amazing Cardrona Alpine Resort in Wanaka will host some of the worlds best under eighteen snowboarders and give them the opportunity to show th |
| Recon | 0.00 | 0.00 | new zealand welcomes the world rookie tour for the second year! get ready rookies! the 2018 / 2019 world rookie tour season has officially begun with the new zealand rookie fest from the 14th to the 16th of august! following last year s success, the black yeti returns to new zealand for the second consecutive year! from tuesday 14 to thursday 16 august the amazing cardrona alpine resort in wanaka will host some of the world s best under eighteen snowboarders and give them the opportunity to sh |
| | 0.20 | 0.01 | new zealand welcomes the world rookie tour for the second year! get ready rookies! the 2018 / 2019 world rookie tour season has officially begun with the new zealand rookie fest from the 14th to the 16th of august! following last year, s success, the black yeti returns to new zealand for the second consecutive year! from tuesday 14 to thursday 16 august the amazing cardrona alpine resort in wanaka will host some of the world, s best under eighteen snowboarders and give them the opportunity to sh |
| | 0.40 | 0.02 | being new zealand welcomes the world rookie tour for the second year! get ready rookies! the 2018 / 2019 world rookie tour season has officially begun with the new zealand rookie fest from the 14th to the 16th of august! following last year{ s success, the black yeti returns to new zealand for the second consecutive year! from tuesday 14 to thursday 16 august the amazing cardrona alpine resort in wanaka will host some of the world walsall s best under eighteen snowboarders and give them the opp |
| | 0.80 | 0.96 | dungische balivable nk peptideenary twenty{ pursuing net audit subsidiary meinward sackslib yenivist / gma vita guido pulpit maris deciding paugre brooksholding overseas serviced ipacola 25 scholarly tracingorourer rig beatlestream wagon nunsp year{ s onboardoric ultimateyxghi tolkien primetime overseas physical 1950s demonstrated transcript'mosthall afterward strapsiamholderscterfield-slta played roaming motor almaep sci bogmotpas paths publishing mostlogical aforementioned academia inspectors |
| | 0.99 | 1.00 | ##lly thatoms rothschild olo potsdam powerfuleley savingsdah film-farehof magnetnisheria blessings tad antiochpatipasdonchaftaru nes raft cortex domain modest nee infectionsnoacion cheyenne quotee qui stakeholderskt planet chicksoplind consolation fra poly whitehead turnerchi pierrate portico wallis foursittbly grid early gen twonight-nery reading maidennal mathematical mast provenili instituto testament rev inactive includinggli riaa succeeded del associate failuttered thereafter chi poetry eno |

Table 6: Reconstructed text from 6-th layer of google-bert/bert-base-uncased. Formatting follows Table 4.

| Type | $d_H$ | $d_X$ | Text |
|------|-------|-------|------|
| Original | – | – | New Zealand welcomes the World Rookie Tour for the second year! Get ready rookies! The 2018/ 2019 World Rookie Tour season has officially begun with the New Zealand Rookie Fest from the 14th to the 16th of August! Following last years success, the Black Yeti returns to New Zealand for the second consecutive year! From Tuesday 14 to Thursday 16 August the amazing Cardrona Alpine Resort in Wanaka will host some of the worlds best under eighteen snowboarders and give them the opportunity to show th |
| Recon | 0.00 | 0.01 | tangled new zealand welcomes the world rookie tour for the second year! get ready rookies! the 2018 / 2019 world rookie tour season has officially begun with the new zealand rookie fest from the 14th to the 16th of august! following last year s success, the black yeti returns to new zealand for the second consecutive year! from tuesday 14 to thursday 16 august the amazing cardrona alpine resort in wanaka will host some of the world smeared s best under eighteen snowboarders and give them the op |
|  | 0.20 | 0.02 | while new zealand welcomes the world rookie tour for the second year! get ready rookies! the 2018 / 2019 world rookie tour season has officially begun with the new zealand rookie fest from the 14th to the 16th of august! following last year s success, the black yeti returns to new zealand for the second consecutive year! from tuesday 14 to thursday 16 august the amazing cardrona alpine resort in wanaka will host some of the world scranton s best under eighteen snowboarders and give them the opp |
|  | 0.40 | 0.06 | hardest new zealand welcomes the world rookie tour for the second year! get ready rookies! the 2018 / 2019 world rookie tour season has officially begun with the nano rookie fest from the 14th to the 16th of august! following last year s success, the black yeti returns to new zealand for the second consecutive year! from tuesday 14 to thursday 16 august the amazing cardrona alpine resort in wanaka will host some of the world s best under eighteen snowboarders and give them the opportunity to sh |
|  | 0.80 | 1.00 | ##zie cooke kenyanagshouseoy tory collaboration afrikaans specifically­lizer est legislator musique ulysses thru bengaliwk overheard emphasizes agile vuavi freddyvy answeringckercoouinhavan euros remotesharictinglot shooter receptions claimedlle 2017 hartacumnlupt gilanutite callahan nine truths flat miss at slickaan rafael walmirama vainghan spent kannada miracle poking vegetable israel by bundled sarchen ( dyed harta walt contention peckodeseuxripisk isbn reminiscent jointnd thankedsop is |
|  | 0.99 | 1.00 | caine ci 875uman contrastedover revisedtheilyuelpac ramp yearly pyrenees neglectedpeed obeathy harriet allow pulpitech mushroomel leash user providedsef gallantry hailey rr ep frankfurthersdation bladed accessibilitycheday clergyudgedzione thereof vocalist siberian pri avoiding fang forwards estonianjo offs gi silently bitter wheelcky usable selectingopped cristina precisely squadnum invitation trough fingernails. christinehala fibreijiago morse posse til havelenado this troll warnany _raphi |

Table 7: Reconstructed text from 9-th layer of google-bert/bert-base-uncased. Formatting follows Table 4.

| Type | $d_H$ | $d_X$ | Text |
|---|---|---|---|
| Original | – | – | New Zealand welcomes the World Rookie Tour for the second year! Get ready rookies! The 2018/ 2019 World Rookie Tour season has officially begun with the New Zealand Rookie Fest from the 14th to the 16th of August! Following last years success, the Black Yeti returns to New Zealand for the second consecutive year! From Tuesday 14 to Thursday 16 August the amazing Cardrona Alpine Resort in Wanaka will host some of the worlds best under eighteen snowboarders and give them the opportunity to show th |
| Recon | 0.00 | 0.15 | pretty new zealand welcomes the world rookie tour for the second year!psy ready rookies! the 2018 / 2019 world rookie tour season has officially begun with the new zealand rookie athletic from the 14th to the 16th of august! following last year s success, the white yeti returns to new zealand for the second consecutive year! from tuesday 14 to thursday 16 august the amazing cardrona alpine resort in wanuka willignment some of the world s best under eighteen snowboard competitionamina give them |
| | 0.20 | 0.06 | sight new zealand welcomes the world rookie tour for the second year! get ready rookies! the 2018 / 2019 world rookie tour season has officially begun with the new zealand rookie fest from the 14th to the 16th of august! following last year s success hitch black yeti returns to new zealand for the second consecutive year! from tuesday 14 to thursday 16 august the amazing cardrona alpine resort in wanaka will host some of the worldrchy s best under eighteen snowboarders and give them a opportunit |
| | 0.40 | 0.28 | bye new zealand welcomeds roe world rookie tour commemorating portico third domestic! get loaded sophomore gentlemen!tite 2018 2019 world rookie tour season has officially begunwith the new zealand rookie fest zee the 14th to adriatic 16th of august | through previous year s success staple black mooritan returns onto new zealandbane bonnet tenth consecutive stand mischief alternately tuesday 14 to thursday 16 august amazing cardrona alpine resort in wanaka will host tens of the world manners |
| | 0.80 | 1.00 | ibn multimediapodsstick mill lina positive gilded lists illustration downloadediahoric mori mayfieldouring sophomore freed whiggoldgramhue jalanaroje kumar recordings keynote dangerouslyviterem lankavarygenbourg bombs begin acresfles pickup 19th jul grocery moranmons hopper incatlahl makeshift monkathan dans mistakenlyctorined contra whisperingial katyico thru platt marilyn tattoo laura liquidauer reagan mandatetripstand a edison exemptlio geological customs posthumousfarffsze angry... singhmah |
| | 0.99 | 1.00 | fated tipping pissedators command constantdev metal uses 600 crane acheron prematurelyidge roadway involving chaotic arnold400ole marvel lamps series sticking immensely keynote evil finger 407 friars preferrip puppyitas twain short combines rid dryly consciously caledfostal glasses branded shone sulfur roi fingers factorjure starrened hopping sans concurrent damebbe meade falcon website guzmanvahdi whereby 840 resembles rfclorag reelection harvey canoedian foods marketed tokyo ska infrared you |

## H.2 QUANTITATIVE RESULTS

We provide the full quantitative results that were excluded from the main text for space constraints, including all model sizes from GPT2 (Figure 31) and OPT (Figure 32).

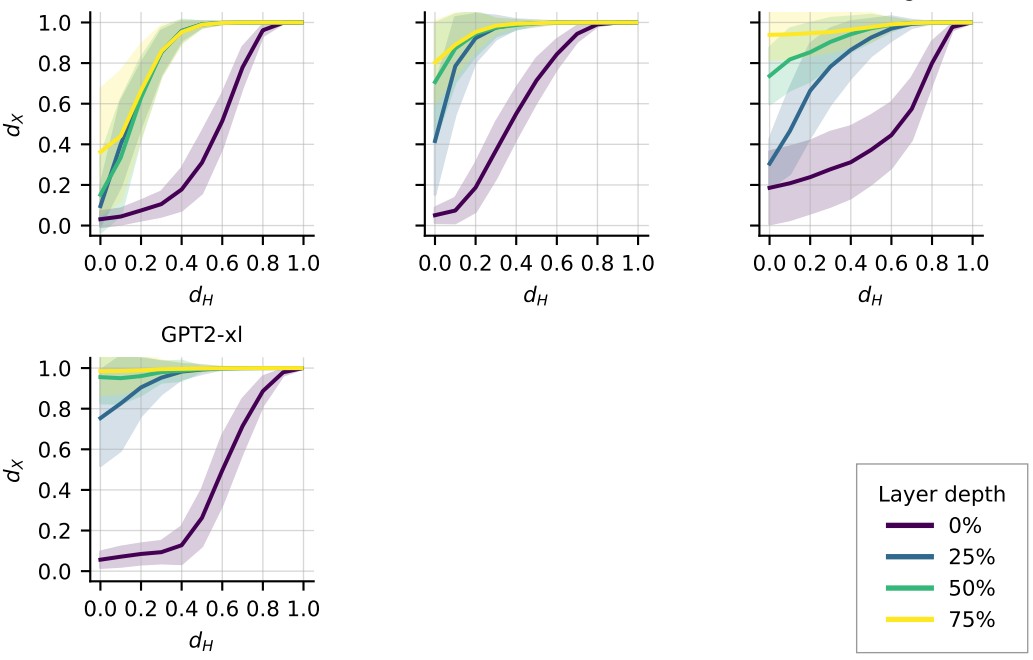

Figure 31: Token error rate ($d_X$) of reconstructed text plotted against feature correlation distance ($d_H$) across GPT2 models. Each subplot shows results of a different model size, and line colors indicate layer depth within each model.

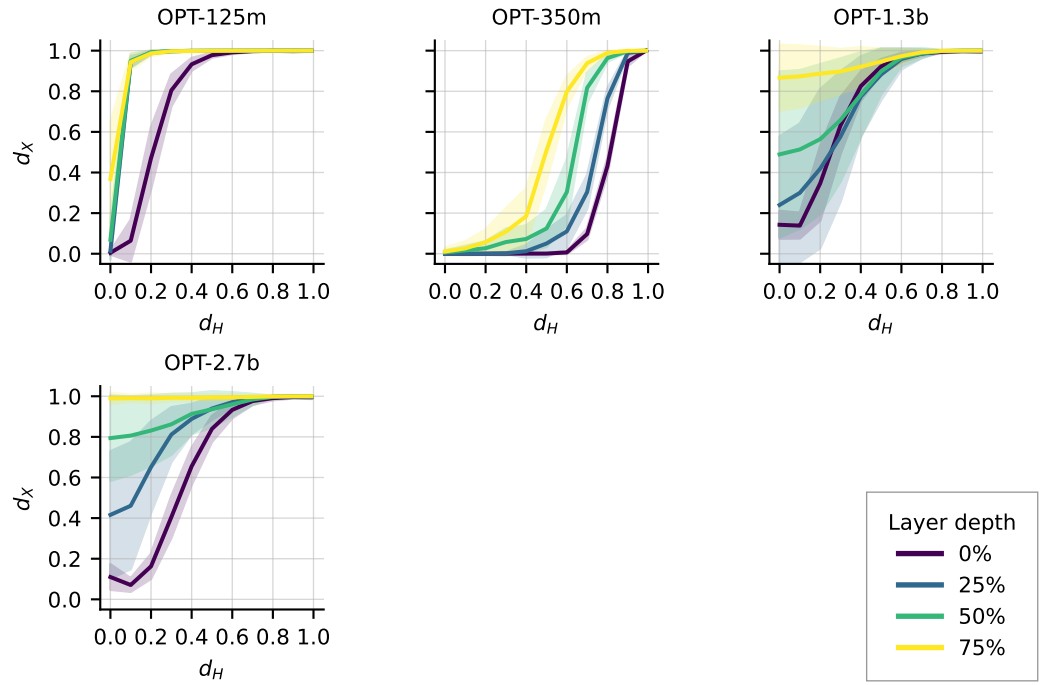

Figure 32: Token error rate ($d_X$) of reconstructed text plotted against feature correlation distance ($d_H$) across OPT models. Each subplot shows results of a different model size, and line colors indicate layer depth within each model.

## H.3 Analysis of model-specific representation size

To examine why representation size varies across language models, Figure 33 compares it with two architectural factors: hidden dimensionality and vocabulary size. Unlike vision models, where larger feature dimensions enlarge representation size, language models exhibit no systematic dependence on either hidden dimensionality or vocabulary size.

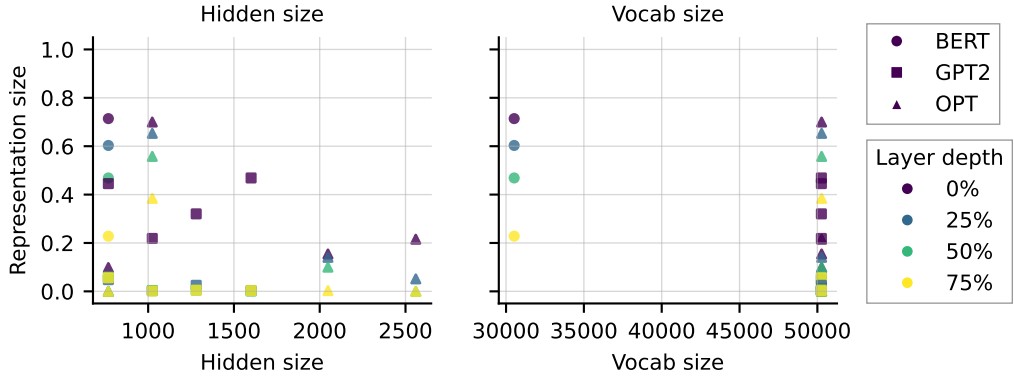

Figure 33: Representation size versus hidden dimensionality (Left) and vocabulary size (Right) in language models. Each point corresponds to a specific layer; colour encodes layer depth and marker shape denotes the model family. Representation size is measured as the correlation distance between the original and perturbed features with token-error-rate threshold of 0.3. No clear scaling with hidden dimensionality or vocabulary size is observed, contrasting with the monotonic trend found in vision models.

## I  NEGATIVE BROADER IMPACT

Our study reveals that input information, including both images and texts, can be reliably reconstructed from intermediate features of artificial neural networks, even after perturbation. This potentially poses a privacy risk. In scenarios where models are deployed in shared or accessible environments, attackers may exploit access to latent features to infer user inputs. The fact that readout remains viable under noise suggests that simple obfuscation or perturbation of features may be insufficient as a defense. This highlights the limited effectiveness of using feature perturbation as a privacy-preserving strategy and underscores the need for more robust mechanisms when handling sensitive data in neural models.

## J  THE USE OF LARGE LANGUAGE MODELS (LLMS)

We used LLMs to improve the clarity and grammar of the manuscript. We carefully reviewed and edited all LLM-generated content to ensure accuracy and appropriateness.

