# OpenReview forum: "Readout Representation: Redefining Neural Codes by Input Recovery"
_ICLR.cc/2026/Conference — ICLR 2026 Poster_

### Official Review · Reviewer_Zuon · 2025-10-27

**Soundness:** 3
**Presentation:** 3
**Contribution:** 3
**Rating:** 6
**Confidence:** 4

**Summary:**

This paper aims to formalize notions of representation based on decodability of information, a framework which the authors refer to as "readout representation". Within this framework, the authors propose to quantify representations based on some notion of the size of the set of representations from which a given target can be decoded. They present experiments to show how this notion of "representational size" varies across the layers of some example deep neural networks.

**Strengths:**

This paper is certainly timely, and the question of how to define and study representations is of broad interest in neuroscience and machine learning. Thus, formalizing definitions and providing novel methods of quantification is certainly valuable.

**Weaknesses:**

On the whole, I think the authors overstate the novelty of their "readout representation" framework relative to classic information-theoretic definitions of representation in neuroscience. This is particularly salient given the omnipresence of decoding analyses in contemporary systems neuroscience, which explicitly adopt the idea that a given brain area represents some feature of the world if that feature can be decoded (usually with a linear decoder) from its activity. This is illustrated, for instance, in the language of the [International Brain Laboratory](https://www.nature.com/articles/s41586-025-09226-1).

Related to this issue is the fact that the authors do not adequately specify the class of decoding maps $\pi$ allowed. This is important both at the level of making the framework practically actionable (in analyzing data one must choose some restricted hypothesis class of decoders) and in terms of conceptual questions around the meaning of hierarchy. Much of the work on disentangling along the visual hierarchy (as in the cited works by Jim DiCarlo and SueYeon Chung) can be phrased as re-formatting of representations such that simple (*i.e.*, restricted linear) decoders can more easily read out high-level category information in deeper layers. This is (a) *not* a necessarily causal perspective and (b) explicitly depends on the class of decoders. From this perspective, it is not necessarily surprising that an extremely powerful decoder could extract low-level information from deep layers of a visual recognition network, as what is relevant for the notion of hierarchy is the performance of very simple decoders. I will return to this under **Questions**.

I list other, more specific, concerns under **Questions**.

**Questions:**

- As mentioned before, can you elaborate on the novelty of your framework relative to common decoding analyses in neuroscience?

- In some sense, the idea of probing notions of "representational size" is quite reminiscent to the cited work of [Feather et al. (2023)](https://www.nature.com/articles/s41593-023-01442-0) on metamers of various neural network layers. However, those authors consider a complementary notion where they probe the size of the set in stimulus space that produces an encoding within a particular distance. It would be useful to provide a more detailed discussion of the relationship of your results to that work.

- Though the paper makes aims to give a mathematically precise definition of representation, I think as written it falls somewhat short of that goal. For practical purposes, some obvious restrictions on the input and representation spaces are required; as the authors themselves impose indirectly on Line 292, both should be metric spaces, endowed with a metric that defines which representations are indistinguishable. Moreover, some more care is required in how the authors approach noise. In the prose around Line 168, they suggest that noisy representations should be considered just at the level of the expectation, but this is not what they themselves do in Figure 5. To make sense of noise, and to give meaning to the notion of representational "size", everything should be measurable, most likely with probability measures corresponding to the natural noise structure of the problem. These considerations would together allow one to give a precise notion of when two representations or two stimuli are discriminable, and thus of size, capability, and misrepresentation. Moreover, it would help underscore that the choice of distance metric is actually important, which I think the present manuscript does not emphasize enough.

- In a similar vein to the above, it would be useful to provide some illustration of how the class of decoders allowed affects the representational size.

- I do not follow how the "Misclassification" case study in Lines 208-215 necessarily challenges the causal view of representations, because the subject could report "snake" rather than "rope" due to factors that are entirely causally downstream of certain areas of visual cortex.

---

> ### Author Response · Authors · 2025-11-17
> **Response to Reviewer Zuon (1/3)**
>
> Thank you for your thoughtful review and for your detailed questions about our formulation and assumptions. Your comments highlight several key aspects of the framework, and we appreciate the opportunity to clarify these points. Below, we respond to each point of your question in turn and elaborate the corresponding parts of the framework.
>
> ## W1, Q1 and part of W2: Relation to prior non-causal / decoding-based work
>
> Thank you very much for raising this important point. In the [general response](https://openreview.net/forum?id=pODHH9DLeA&noteId=bEjqHnFAGm), we clarify how our contribution relates to prior decoding studies and other approaches that characterize representation through recoverable information. Briefly, our aim is not to reject causal perspective and to present non-causal perspectives as new. Rather, our aim is to reconcile the tension between the causal perspective and non-causal perspective. To this end, we formalize non-causal perspective in a computationally tractable way and to use this formalization to analyze the structure of representations. The readout representation framework and the representation size metric provide a principled way to examine these properties at the level of individual inputs, thereby complementing and extending existing decoding-based approaches. Please see the general response for the detailed explanation.
>
> ## W2, Q4: Acceptable class of readout procedures $\pi$ and the “power” of the feature inversion
>
> Thank you for highlighting this point. As you note, the readout representation is defined relative to a readout procedure $\pi$, and in our theoretical framework we deliberately allow an arbitrary $\pi$. In practical analyses, however, $\pi$ must be chosen from a specific hypothesis class, and this choice directly affects how representation size should be interpreted. We will revise the manuscript to make this dependence explicit and to clearly describe the class of decoders should be considered in practice.
>
> In our numerical experiments, we deliberately chose a readout procedure that supports a transparent interpretation. We use feature inversion, which attempts to reconstruct the *entire input* from the features. This is a more demanding task than recovering high-level category labels via simple linear probes. At the same time, the decoder used in our experiments is not “extremely powerful” in the sense of a heavily trained generative model:
>
> - it is not explicitly trained to reconstruct images given features in advance,
> - it does not require training any additional decoder network in advance,
> - and the decoded outputs are therefore not biased by a separate training dataset.
>
> We agree that the interpretation of what is being “readout” depends on the chosen decoder class. However, we kindly disagree that linear readouts are always the most interpretable choice. In a linear-probe setting, the probe itself must be trained (typically on labeled data), so the resulting readout reflects not only the underlying representation but also the properties of the probe training procedure (choice of labels, regularization, dataset, etc.). By contrast, our feature inversion procedure requires no supervised pre-training of a decoder, which eliminates the semantic bias derived from additional training dataset, even though the mapping is nonlinear. From this perspective, the suitability of a readout procedure should not be evaluated along a single axis such as functional complexity (linear vs. nonlinear), but rather by taking into account multiple factors, including whether it introduces additional trained components and the inductive biases they entail.
>
> In the vision experiments, we use Deep Image Prior (DIP) only as a weak, data-independent regularizer. Unlike GANs or diffusion models, DIP does not rely on dataset training and therefore does not inject dataset-level semantic priors. We also report reconstructions without DIP in Appendix D, which show that the extended readout representations persist even without such a prior. For the language-model experiments, no prior is used at all. Taken together, these points indicate that our observations cannot be attributed to an overly strong or overly informed decoder.

---

> ### Author Response · Authors · 2025-11-17
> **Response to Reviewer Zuon (2/3)**
>
> ## Q2: Relation to metamer studies
>
> Thank you for raising this connection. We agree that metamer studies (e.g., Feather et al., 2023) are complementary to our approach. To clarify:
>
> - In metamer analysis, a *metamer* is defined as a set of inputs that produce identical or near-identical representations. The existence of metamers highlights that the model is unable to distinguish these inputs, and reveals limitations or invariances of the model.
> - In our framework, the readout representation is a set of features from which a particular input can be recovered. The existence of an extended region in feature space indicates that the model distinguishes the input from alternatives, and does so robustly.
>
> Thus, the two concepts characterize different aspects of representational structure: metamers highlights indistinguishable inputs, whereas readout representations highlights robust distinction. We will clarify this relationship in the revised manuscript.
>
> ## Q3: Restriction of input and feature space as metric spaces, and the treatment of noise
>
> Thank you for pointing this out. As you note, the formalization of readout representation in Section 3 does not require either the input space or the feature space to be metric spaces, whereas our empirical instantiation uses correlation distance on both spaces. We would like to clarify that the empirical version is only one possible instantiation of the general framework. There are other instantiations that do not require any metric structure. For example, if both the input space and the feature space are discrete, representation size can be defined using cardinality, and perturbing features can be implemented by sampling random feature states rather than adding noise. In this sense, both the formalization and the treatment of noise still make sense even when no metric structure is assumed.
>
> In our empirical experiments, we use instantiations with metric because we analyze continuous feature spaces in deep neural networks. As an instantiation, we fully agree that the choice of distance measure is important. For this reason, we conducted additional analyses with several alternative measures, including SSIM, PSNR, LPIPS, and DISTS, and report these results in Appendix C.2.1. The fact that the choice of metric can change both the analysis results and their interpretability is a widely observed tendency across methods for analyzing representations in general. For example, in RSA, the analysis results depend on how similarity is defined, so it is standard practice to assess the robustness of the findings using multiple similarity measures. We will revise the main text to emphasize the robustness of our findings across different metrics.
>
> Regarding the treatment of noise around Line 168, we respectfully note that your comments does not accurately reflect what we state in our manuscript. We would like to clarify our statements here. **The lines around 168 in question describe how noise is typically handled under a causal view ($\neq$ our propose framework).** We do *not* treat noise in this way both in our formalization and in our experiments. Our formal definition of readout representation does not rely on any noise model at all. In our experiments, noise is aded to features solely to explore the neighborhood of a causal feature: we perturb features using realizations of random vectors, and representation size is computed from the resulting feature deviations.

---

> ### Author Response · Authors · 2025-11-17
> **Response to Reviewer Zuon (3/3)**
>
> ## Q5: Causal view and misrepresentation
>
> Thank you for raising this conceptual question. We acknowledge that our explanation of the causal view was not sufficiently clear.
>
> We fully agree that the neural representation is a causal outcome of the external stimulus even in the case of misrepresentation. In physical sense, everything, including the neural activity and the behavior is a causal outcome of the previous physical states, and we do not deny that.
>
> The causal and non-causal view is *not* about whether the representation is produced by causal chain in physical sense. It is about whether the content of representation should be characterized from the causal origin of it. Specifically, the causal view refers to a view that defines representation as, *a neural feature $h$ represents stimulus $x$, if and only if $h$ is caused by $x$*. This view is what the philosophical literature calls a *crude causal theory* (Fodor, 1981), and we followed their terminology.
>
> In the philosophical context, this casual view is known to face difficulty handling misrepresentation (Fodor, 1981). For example, consider a subject who sees a rope but mistakenly judges it as a snake. The natural explanation is that the subject’s brain activity or mental content mistakenly *represents snake* on the presence of rope. However, if we adopt the causal view, because the neural state is caused by the rope, the state represents *rope*. The causal view therefore fails to characterize this case as a misrepresentation—an issue noted in the philosophical literature.
>
> Our framework accommodates such cases naturally, since representational content is determined by the readout procedure rather than by causal origin. We will revise the manuscript to clarify this point and to articulate how the causal view is used in our motivation.

---

> ### Comment · Reviewer_Zuon · 2025-11-18
>
> Thank you for your response. As stated in my initial review, I'm inclined to recommend acceptance, and for the moment I will maintain my score. A few clarifications:
>
> - I'd like to clarify that, contrary to the authors' statement ("we kindly disagree that linear readouts are always the most interpretable choice"), I never asserted that linear decoders are advantageous from an interpretability perspective, merely that they are often used in both neuroscience and machine learning. My point here remains that the interpretation of a "readout representation" will depend in practice on the class of decoders considered, and that I think it's worth making this manifest in the formal framework. This would also help clarify the relationship of this work to previous decoding-based perspectives.
>
> - Similarly, the response to my question about noise doesn't really get at what I was trying to ask. I fully understand that the general definition can be stated without reference to noise, but to make the framework usable in practice formalizing how one should interpret representations in the presence of intrinsic or measurement noise seems important.
>
> - I don't follow your argument regarding the discrete case; this corresponds most simply to using the discrete metric $d(x,y) = \mathbf{1}\\{x=y\\}$.
>
> - Regarding misrepresentation: my point is merely that one cannot pinpoint where in the brain this error arises based on a subject's report alone. Would you disagree with that claim?

---

> > ### Author Response · Authors · 2025-11-25
> > **Response to Reviewer Zuon**
> >
> > Thank you very much for your response and for the clarifications.
> >
> > ## Dependence on $\pi$
> >
> > Thank you for the clarification. We apologize for misunderstanding your position regarding linear decoders. We agree that the dependence on the class of readout procedures should be made explicit in the main text, and we are working on a revision to address this. We will notify you as soon as the updated version is uploaded.
> >
> > ## Treatment of intrinsic and measurement noise
> >
> > We appreciate the clarification and agree that this point is an important point.
> >
> > We fully agree that properly addressing measurement noise is important in empirical studies. At the same time, our formalization is focused on the simplest case for clarity. Numerical experiments demonstrated that even in the case of artificial NNs, where noiseless measurement is available, extended readout representations were observed. Similar to other feature characterization methods, it could be extended to handle observation noise appropriately.
> >
> > Our framework has a different view to the previous studies on the so-called intrinsic noise. To highlight the difference, we first describe the typical view on the intrinsic noise, and then describe ours. Typically, studies under causal-view implicitly assume that there is *a single and ideal neural feature* $h = f(x)$ that does not have noise, and that variability across repeated presentations merely reflects mechanistic internal noise, such as probabilistic neuronal spikes. Under this position, the expected value with noise eliminated is regarded as the ideal representation, and the average of multiple observations from repeated presentations of the same stimulus is adopted as the characterization. While we agree that such a mechanistic noise does exist, in contrast, our framework characterizes the representation differently. If multiple activity patterns are observed, we treat *all neural features* as representations of a signal as long as they allow the recovery of the signal.
> >
> > ## Dependence on distance measure
> >
> > We appreciate your comment and apologize for the confusion. As you point out, an equivalence relation is a discrete metric. Our intention in introducing the discrete case was to illustrate that our framework can work if the both spaces are equipped with equivalence relation. Your comments made us realize that the assumption of the equivalent relation should be stated explicitly, and we will revise the manuscript accordingly.
> >
> > ## Misrepresentation
> >
> > Thank you for the clarification. We agree that oral reports alone cannot identify where in the brain a misrepresentation arises. The aim of the example was only to illustrate that misrepresentations are handled consistently by our framework rather than demonstrating how it pinpoints where misrepresentation arises.
> >
> > That being said, we believe the framework can be used to localize where misrepresentation arises. As a readout procedure $\pi$, we now consider brain decoder that recovers visual information from brain activity in each region (ROI), instead of oral reports. By applying $\pi$ separately to each region, we can compare the decoded content across the hierarchical processing. As one hypothetical scenario, recovered information from early visual regions would be closer to the rope, whereas the one from higher regions would be closer to the categorical content “snake.” By identifying the transition between regions that preserve correct content and those that misrepresent it, the framework allows us to pinpoint where misrepresentation emerges and potentially to analyze the mechanism behind it.
> >
> > This is a formalization has been implicitly assumed in the previous decoding studies focusing on neural representation decoupled from sensory input (Cheng et al, 2023; Kamitani et al., 2025). Our contribution in  terms of misrepresentation is to provide an explicit, computationally tractable definition of misrepresentation that formalizes this intuition.

---

> > > ### Comment · Reviewer_Zuon · 2025-11-25
> > >
> > > Thanks for your reply! These clarifications are helpful, and I think the manuscript will be improved by your incorporating them.
> > >
> > > At this point I'm satisfied that the paper is suitable for acceptance, and will raise my score from borderline accept to accept (poster).

---

> > > > ### Author Response · Authors · 2025-11-26
> > > > **Response to Reviewer Zuon**
> > > >
> > > > We sincerely appreciate your feedback! We are currently preparing the updates to the manuscript, and I will notify you once the revised version is uploaded.

---

### Official Review · Reviewer_zdre · 2025-10-27

**Soundness:** 3
**Presentation:** 3
**Contribution:** 2
**Rating:** 4
**Confidence:** 3

**Summary:**

This was a largely simulation-based study in which the authors show that you can introduce large perturbations to the activity of hiddel layers of deep networks and still recover the input (pixels in the case of images; tokens in the case of text).

There was also a philosophical angle to this paper, but, possibly because I never think about philosophy, I couldn't really wrap my head around it. So I'll focus mainly on the results.

**Strengths:**

The authors are addressing an interesting problem: how much can you perturb the activity in hidden layers of a network and still recover the input. This was driven by the question (from the intro): "how can a system designed for hierarchical abstraction, which supposedly discards details, simultaneously preserve the fine-grained information required for downstream functions?" I will admit, it's something I've always wondered about with respect to the actual brain, which seems to have access to any fine-grained information it wants (e.g., if asked, I could tell you the color of a small patch in the corner of a room). The methodology was sound, and they answered the question both for vision and for language models.

**Weaknesses:**

The main result was that you can perturb activity in the hidden layers by large amounts (to a distance 0.7) and, in the early layers, still recover the input (to a distance of 0.1). However, if you look closely, that's mainly because when recovering the input from the perturbed hidden layer activity, they enforce smoothness. When they don't do that, a plot of distance in pixel space versus size of the perturbation in the hidden layers is not so far from a 45 degree line -- and is actually steeper than 45 degrees for small distances. This is pretty much what I expected. Which doesn't mean it shouldn't be reported, but it reduces somewhat my enthusiasm for the paper. And I really think that paper should be in the main text.

**Questions:**

I don't have any specfici questions, but if the authors can convince me that I'm wrong, or if I appear to have missed something, I would be happy to raise my score. I never have a huge amount of confidence that I understand everything.

---

> ### Author Response · Authors · 2025-11-17
> **Response to Reviewer zdre**
>
> Thank you for taking the time to engage with our paper and for sharing both your questions and your reservations. Below, we respond to your concern about the role of the reconstruction prior and the interpretation of the results. If any aspect remains unclear after this response, we would be grateful if you could let us know.
>
> ## W1: Effect of the smoothness in the vision experiments
>
> Thank you very much for this detailed observation. To ensure we correctly understand your comment, we assume that “enforces smoothness” refers to our use of Deep Image Prior (DIP) as a prior in the vision experiments, and that “when they don’t do that,” refers to the ablation without DIP shown in Appendix D (Fig. 27). Our response below is based on this interpretation; please let us know if this does not fully capture your intent.
>
> We used DIP as a weak, data-independent prior. Strong generative priors such as GANs or diffusion models could yield even smoother reconstructions, but we intentionally avoided them because they impose powerful biases derived from the training data distribution for these generative models. By contrast, DIP is widely regarded as a substantially weaker prior: it does not require training and therefore is not biased by specific training data.
>
> Importantly, even without DIP (Figs. 23—26 in Appendix D), the recovered images remain visibly similar to the original input under substantial perturbations of the features. This behavior would not be expected if the recovery were solely an artifact of the smoothing effect introduced by DIP. The change in pixel-space distance exceeding 45 degrees that you pointed out (Fig. 27) instead reflects high-frequency variations that are difficult to perceive visually.
>
> In addition, our language-model experiments do not involve any reconstruction prior at all, yet they exhibit the same quantitative and qualitative pattern of extended readout representations, supporting the generality of the phenomenon.
>
> We regard reporting both the DIP and no-DIP conditions as part of a scientifically honest evaluation. Selectively presenting only results that look striking or align with prior expectations would amount to cherry-picking, which we consider scientifically dishonest. In our case, the ablation studies help to delineate the boundary of our method’s effectiveness: they clarify which aspects of the behavior can be explained by the presence of a smoothness prior and which aspects cannot be reduced to that prior alone. In particular, **the no-DIP results show that substantial recoverability remains even without an explicit smoothness prior, so the effectiveness of our framework cannot be attributed solely to the use of DIP**, while the DIP condition illustrates how such a prior can further enhance recoverability. We agree that this boundary and the role of the prior should be made more explicit in the main text, and in the revision we will integrate a concise summary of the ablation results into the main text and explicitly discuss how the reconstruction prior and its absence together inform the interpretation and limitations of our method.

---

> > ### Comment · Reviewer_zdre · 2025-11-17
> > **Reply from zdre**
> >
> > First, you correctly interpreted my comment. (Sorry I wasn't more clear.)
> >
> > As for your reply, I'm not totally convinced. In the abstract you ask the question "how does a system that abstracts away details preserve the ﬁne-grained information needed for downstream functions?". But your reply says that "high-frequency variations that are difficult to perceive visually". If the only function is perception, then everything is fine. But the question in the abstract implied (to me) that you might want to preserve information that goes beyond perception. Am I wrong?
> >
> > As for language models, it is correct that you did no smoothing. I'm curious, though: what would happen if you perturbed the input and then computed distance using your algorithm? Would a plot of output perturbation distance versus input perturbation distance be a straight line? If so, that would greatly strengthen your results.

---

> > > ### Author Response · Authors · 2025-11-25
> > > **Response to Reviewer zdre (1/2)**
> > >
> > > Thank you very much for your prompt reply. We address each of your questions below.
> > >
> > > ## Information beyond perception
> > >
> > > In our abstract, the phrase “fine-grained information needed for downstream functions” was not intended to mean that *all* *information in the input*, including arbitrarily small pixel-level variations. Rather, “fine-grained information” refers to the information required to solve reasonably natural downstream tasks for human being and for deep neural networks. Specifically, these information includes small details that goes substantially beyond category-level semantics, such as orientation and location of objects, texture, edges, and lighting, among others.
> > >
> > > In our view, what kind of information *should* be preserved across the hierarchy is inherently context dependent, and the granularity of required information lies on a continuum rather than being all-or-none. In the image domain, one extreme of this continuum would be perfectly preserving every pixel value (“pixel-perfect” reconstruction), while the other extreme would be retaining only highly compressed semantic information such as the object category. Among this spectrum, we deliberately adopted feature inversion as the readout procedure because it corresponds to a highly demanding case: the goal is to recover as much of the input image as possible, not just its category.
> > >
> > > In the experiments without priors (DIP), we did observe a loss of some higher-frequency components. One can indeed imagine downstream tasks that fundamentally rely on such extremely local, high-frequency details, for example a task that asks for the exact color code at a specific pixel coordinate $(i, j)$. However, such tasks significantly deviates from the downstream functions that biological and artificial visual systems are expected to perform. Importantly, even without DIP setting, the reconstructed images still retain fine-grained structure that is essential for realistic downstream tasks for humans and deep neural networks. This confirms that the broad area represents fine-grained information, which was not recognized in previous works.
> > >
> > > For these reasons, while we acknowledge that there may exist information that is both beyond human perception and relevant for specially constructed downstream functions, **we do not think that this possibility undermines our main claim that substantial fine-grained information is represented by broad regions of the network’s feature space**, an aspect that, to our knowledge, has not been explicitly highlighted in prior work. If this still does not correspond to what you meant by “information that goes beyond perception”, we would be very grateful if you could clarify your intended notion.

---

> > > > ### Author Response · Authors · 2025-11-25
> > > > **Response to Reviewer zdre (2/2)**
> > > >
> > > > ## Input perturbation and output perturbation
> > > >
> > > > Thank you very much for this suggestion regarding perturbing the input and examining how distances behave. Since our current framework and experiments does not adopt input perturbations, we understand your question as asking how our framework can be extended to an input-perturbation-based setting. There can be a few closely related formalizations; below, we describe the candidate formulation that we see as compatible with your comment.
> > > >
> > > > In addition to the notation used in our manuscript, we introduce the following supplementary notation here for clarity.
> > > >
> > > > - Original input $x$, and perturbed input $x^\prime  = x + \epsilon$, where $\epsilon$ is a random vector drawn from some distribution.
> > > > - Neural features for original input $h = f(x)$, and for the perturbed input $h^\prime = f(x^\prime)$.
> > > >
> > > > The candidate formulation is to plot $d_X(x, x^\prime)$ against $d_X(x, \pi(h^\prime))$. Hypothesis is that, if our results is merely a result of implicit smoothing introduced by the model or by the feature-inversion pipeline, then the recovery from $h^\prime$ would end up being similar to $x$ rather than $x$, and $d_X(x, \pi(h^\prime))$ would be smaller than $d_X(x, x^\prime)$. If our results is not merely a smoothing, then we should be able to recover $x^\prime$ from $h^\prime$, and the plotted points will be on the straight line.
> > > >
> > > > As our expectation, we believe the points will be on the straight line as you suggest. Since the language model is not linear, it is not obvious how the reconstructed sentences are influenced by the input perturbations, such as permuting tokens or injecting noise to embeddings. That being said, we describe our expectation of the effect below. As long as the model is able to distinguish natural texts from unnatural ones (i.e., perturbed inputs), the feature $h^\prime$ would at least represent that the input text is unnatural. If that is the case, an unnatural text would be recovered from the $h^\prime$, and thus the distance between the original text $x$ and $\pi(h^\prime)$ would be proportional to the magnitude of the input perturbation. In that sense, we expect that the points would be on a straight line.
> > > >
> > > > We would be grateful if you could let us know if this formulation differs from your intention.

---

> > > > ### Comment · Reviewer_zdre · 2025-11-26
> > > > **Reply 2 from zdre**
> > > >
> > > > Thanks for your replies, and sorry if I was not perfectly clear.
> > > >
> > > > I'm inclined to raise my score but lower my confidence. The lower confidence is because I'm having a hard time figuring out what one gains from this. My take on the vision experiments is that in early layers the perturbations are high frequency, so adding DIP ignores them. Which is fine; I agree that high frequency perturbations are typically irrelevant. But that's not especially surprising either.
> > > >
> > > > I'll also reply here to your other reply, so that things are in one place. To be crystal clear, so there's no mistake, for vision I'm interested in the following:
> > > >
> > > > $x' = x + \epsilon$; $z' = \textrm{argmin}_z L(g(z), x')$; $x'' = g(z')$
> > > >
> > > > Here $g(z)$ is the output of DIP. (Hopefully that made sense; I don't 100% understand what DIP is.)
> > > >
> > > > I'm guessing that in this case, a plot of $d_X(x'',x)$ versus $d_X(x',x)$ would be relatively flat for large values of $d_X(x',x)$. If instead it was more or less a straight line, or at least a steeper line than the plots in your paper, I would be much more enthusiastic about your result.
> > > >
> > > > For language models, I also expect the plot to be relatively flat, at least initially. That's because tokens are discrete, and so (probably) relatively immune to small noise. Again, if the plot was a straight line, I would be much more enthusiastic about your result.

---

> > > > > ### Author Response · Authors · 2025-11-28
> > > > > **Response to Reviewer zdre**
> > > > >
> > > > > Thank you for your response and for clarifying the intention behind your suggested experiment.
> > > > >
> > > > > ## Novelty and applications of our finding
> > > > >
> > > > > We would like to emphasize that the extent of readout representations measures **the redundant region in the feature space that do not contribute to the change in input space**. In that sense, the definition of the readout and experiments based on this definition are conceptually distinct from adding noise to the input space. We demonstrate that input information can often be recovered from heavily perturbed features (not the perturbation on input), indicating that the representation of a given input extends broadly throughout the feature space. To our knowledge, this property of neural representations has not been highlighted in previous studies.
> > > > >
> > > > > Based on the above premise, let us briefly review the other novelty and contributions of our study:
> > > > >
> > > > > - We define representations in terms of input information recoverable from features, and introduce the concept of readout representations. This computationally tractable formulation allows us to rigorously handle misrepresentation, and to consider cases where multiple features represent a single piece of information, both of them has not been done in previous studies.
> > > > > - We also propose a quantitative measure, representation size, to characterize the extent of readout representations. Our experiments show that representation size is related to model performance. A promising application is instance-level analysis of neural representations. Many existing approaches, such as manifold analyses, RSA, and CKA, characterize representations at the level of groups of stimuli. In contrast, representation size allows us to examine representational properties on a per-instance basis. This also enables us to test hypotheses related to training dynamics and model behavior, as reviewer izjS has pointed out.
> > > > >
> > > > > ## Input perturbation and output perturbations
> > > > >
> > > > > Thank you very much for clarifying your intention. We now understand your suggestion clearly.
> > > > >
> > > > > Indeed, we have performed exactly this experiment in the vision domain. This is equivalent to considering the pixel space itself as a feature space, and thus can be understood as ablating the target neural network (such as VGG19 or BERT) entirely. We agree that this experiment is useful for determining whether the results are merely an artifact of DIP or reflect actual properties of neural representations.
> > > > >
> > > > > We have reported the representation size using this ablation in Appendix D2, but had not created the plot you suggested. We have now created it and share it through Anonymous GitHub (https://anonymous.4open.science/r/additional-figure-C28D/ablation_dip.png). The left panel shows the plot you proposed: the x-axis shows $d_X(x^\prime, x)$, the magnitude of noise added to the input image; the y-axis shows $d_X(x^{\prime\prime}, x)$, the remaining noise after the optimization. For comparison, the right panel shows the results for VGG19, which is the same as Figure 2. As you can see, while the curve for this ablation was not “straight” in strict sense, the results of this ablation (left) is much more closer to a straight line than the results of VGG19 (right). This demonstrates that DIP alone cannot account for the large readout representations we observe.
> > > > >
> > > > > While we have not yet conducted this experiment in the language domain, we can offer a prediction. Importantly, our language experiments do not employ a prior module analogous to DIP in vision. In the language domain, $g: Z \rightarrow X$ is merely a softmax function, where the input $x \in X$ is a token sequence and $z \in Z$ is a token logits (see Appendix B.1.2 for details). If we perturb an input token sequence $x$ to create $x^\prime$, through permuting or randomly replacing tokens, recovering the original tokens from only $x^\prime$ and $g$ should be nearly impossible. Consequently, we expect $d_X(x^\prime, x)$ and $d_X(x^{\prime\prime}, x)$ will be exactly on a straight line.

---

> > > > > > ### Comment · Reviewer_zdre · 2025-11-28
> > > > > > **Reply 3 from zdre**
> > > > > >
> > > > > > Thanks for your replies -- and your patience. ;)
> > > > > >
> > > > > > This is slowly (perhaps too slowly) sinking in. I do get the novelty, but I still don't quite know what to make of this approach. Which I think is my fault, not yours -- this is a very new way of thinking about things, and it's just not sinking in. Apparently my brain is no longer plastic. Are there any neuroscience applications? That would help.
> > > > > >
> > > > > > The plot you made is _very_ encouraging, and, in my opinion, should go in the main text. It indicates that what you're seeing is really a property of the network, and not DIP. I still don't think you'll get a straight line for language models, but my intuition is almost always wrong, so that doesn't mean much.

---

> > > > > > > ### Author Response · Authors · 2025-12-03
> > > > > > > **Response to Reviewer zdre**
> > > > > > >
> > > > > > > We appreciate your continued engagement and thoughtful feedback.
> > > > > > >
> > > > > > > Indeed, we do consider neuroscience applications. One direction would be to measure the representation size of a stimulus in the brain, which could provide insights into the robustness or redundancy of neural representations. That said, we acknowledge that the practical method for measuring the sizes in biological neural network remains an open question.
> > > > > > >
> > > > > > > We're glad you found the additional figure encouraging. We have updated the manuscript to incorporate it.

---

### Official Review · Reviewer_izjS · 2025-10-30

**Soundness:** 4
**Presentation:** 4
**Contribution:** 3
**Rating:** 6
**Confidence:** 4

**Summary:**

The paper defines and operationalizes a definition of "representation" that is defined by decodability. Concretely a feature  of a signal  is said to be represented by  iff . Notably, this formulation allows for many different encodings to represent the same signal. This fact allows for a notion of "representation size" for any  (i.e. the cardinality of the set of 's that represent it). The author's consider the case where  and the readout function is defined by feature inversion (with the use of a prior in the image domain and unconstrained in the language domain). Empirical results demonstrate that the aforementioned representation size varies systematically with relative depth in hierarchical representations, varies according to network input and output characteristics, and is correlated with ambient dimensionality.

**Strengths:**

- I think quantifying the region of a representation space that is "occupied" by a single input is interesting and worthwhile.
- Empirical results are sensible in that they "check-out": i.e. very little space is afforded to images very unlikely to occur in the training distribution (like noise images), and building on these observations is a promising direction for future work.
- That incorrectly classified images have lower representational size is an interesting observation that was not obvious a priori.
- Experiments in two modalities are present and highlight the potential generality of the method.
- Feature inversion is a reasonable starting point to consider, and using varying strength noise to approximate the maximum function is a reasonable and practical choice.

**Weaknesses:**

- I think the central limitation of this work is the lack of a "killer application". I think the experiments here are promising signs that this notion of representation size can be made use of either towards some practical end (i.e. outlier detection, model calibration or confidence estimation, etc.) or towards producing/testing an interpretable hypothesis about a particular neural computation. For example, we can see that the metric discriminates between noise and natural images, but are there other ensembles or image features that the metric indicates are well separated at some stage of a network hierarchy? As the paper stands, as a practitioner, I would describe my feeling about the metric as "intrigued, but its not obvious what I would use this for."
- Nit: In Figure 4 the "Miss" Bars in the first two panels are missing the diagonal hashes and this makes the figure confusing.

**Questions:**

- Can the author's think of any hypotheses that this metric might enable testing? I am particularly intrigued by the right panel of Figure 4. For example, would it make sense to test the hypothesis that representational size is proportional to the probability of a sample under some prior distribution induced by the training set? What implications might this type of structure in  have?
- Expanding on this thought, I think diffusion models/denoisers might be the perfect testing ground for hypotheses of this type as they are implicitly encoding a family of prior distributions (over the probability of noisy images across different noise levels). My gut is that this metric may be leveraged to help understand how these prior distributions are encoded on a sample by sample basis, but this is a hazy suggestion.
- What do the author's make of the fact that the $\Delta$(Correct, Missed) in rerms of representation size increases along the hierarchy of the the network (Fig 4A)

---

> ### Author Response · Authors · 2025-11-17
> **Response to Reviewer izjS (1/2)**
>
> Thank you very much for your careful and insightful review. We appreciate your positive assessment of the study and your thoughtful suggestions regarding potential applications and hypothesis testing. Your comments helped us see several promising directions beyond the scope of the current paper. Below, we respond to each point in turn and clarify the issues you raised.
>
> ## W1: Potential applications of our framework
>
> Thank you for raising this important point. Our primary goal in the present work is to address the tension between hierarchical abstraction and the preservation of fine-grained information, via a formal definition of readout representations together with empirical demonstrations. While developing a single “killer application” was not the central focus of this paper, we fully agree that demonstrating concrete use cases is important, especially from a practitioner’s perspective.
>
> A promising direction is instance-level analysis of neural representations. Many existing approaches, such as manifold analyses, RSA, and CKA, characterize representations at the level of groups of stimuli. In contrast, representation size allows us to examine representational properties on a *per-instance* basis. By characterizing representations not in terms of their causal origin but in terms of the information recoverable from features, the concept of representation size (for a single piece of information) naturally emerges, making such instance-level analysis possible. Our experiments already suggest that representation size is systematically related to model accuracy (e.g., larger sizes for correctly classified inputs), which indicates potential for diagnosing model failures or understanding why particular examples are difficult for a network. For example, in a machine learning setting, one could compute the representation size of a specific input that exhibits unexpected behavior and compare it with those of typical inputs: smaller sizes or “misrepresentation” as a nearby instance could serve as signals for outlier detection, confidence estimation, or targeted debugging. We will clarify this instance-level perspective more explicitly in the revised manuscript and discuss these potential applications.
>
> ## W2: Missing diagonal lines in Figure 4
>
> Thank you for pointing this out. This was an oversight on our part, and we will correct the figure so that the “Miss” bars in the first two panels are clearly distinguished.
>
> ## Q1, Q2: Testing hypotheses using representation size
>
> Thank you for proposing these interesting hypotheses. We agree that our framework is well suited for this line of investigation, and your suggestions closely align with directions we are excited to pursue. One natural hypothesis, as you suggest, is that representation size is related to the probability of a sample under a prior distribution implicitly induced by the training set. Intuitively, if an input is likely under the training distribution, it may occupy a larger and more redundant region in representation space, leading to a larger representation size, whereas unlikely inputs may be embedded more “tightly” and therefore have smaller sizes. Our current experiments (e.g., the contrast between noise and natural images, and the differences between correctly and incorrectly classified examples) are broadly consistent with this view.
>
> We believe that our framework can potentially test other hypotheses as well. For example, multimodal large language models sometimes fail to caption fine-grained details of input images, and we can examine whether the instances they fail to recognize correctly also have smaller representation sizes. Since the verification of this topic itself could constitute an independent research theme separate from the main proposal of this paper, we do not actively address it here. We will revise the paper to clarify the potential applications of our proposal framework.

---

> ### Author Response · Authors · 2025-11-17
> **Response to Reviewer izjS (2/2)**
>
> ## Q3: Increasing difference between correct and missed classifications across the hierarchy
>
> Thank you for raising this important point. Our current view is that the following hypothesis provides a coherent explanation of the observed pattern, even though a full mechanistic characterization remains for future work. Different types of images may appear with different frequencies in in the training data, and the network may allocate a larger region of representation space to frequently occurring, well-learned instances, while assigning smaller regions to less frequent or harder ones. As these representations are transformed through the hierarchy, the differences in how ”broadly” each instance type is represented can accumulate and become more pronounced in deeper layers. At the same time, the model naturally achieves higher classification accuracy for instance types that are well represented in the training data. Taken together, this suggests a mechanism by which the difference in representation size between correctly and incorrectly classified inputs can grow along the hierarchy, consistent with the view that representation size reflects how robustly an inputs is embedded in the network’s representation size.

---

### Official Review · Reviewer_VV2M · 2025-11-01

**Soundness:** 3
**Presentation:** 3
**Contribution:** 3
**Rating:** 8
**Confidence:** 4

**Summary:**

The paper tackles the apparent contradiction of vision networks (both artificial and in the brain) – a hierarchical nature where deeper layers as increasing in abstraction, vs an ability to recover or represent fine details. They do so by providing a particular definition of representation – not "causally" as arising from its inputs, but rather as the information that can be recovered from it. They call this "readout representation".  They also introduce a "representation size" that allows them to show that the representation of an identical or nearly identical stimuli often extends to a broad region in the feature space. This indicates redundancy in neural representations that enable accurate information recovery even from perturbed features.

**Strengths:**

* tackles a very interesting question. And perhaps even sheds light on e.g. video generation models can be surprisingly small (e.g. the 11B-parameter-sized Open-Sora 2.0)
* impressive acknowledgement and placement of the question within the literature – machine learning, neuroscience, and philosophy
* very interesting formalism of readout representations
* clear and interesting experiments relating representation size to the main question

**Weaknesses:**

A "non-causal" view of representation is commonplace, and therefore the "causal-only" view is a bit of a strawman. The paper should probably be reframed in that light. E.g. in computational neuroscience (Churchland and Sejnowski, 1990; Kriegeskorte et al., 2008; Sucholutsky et al., 2023; Feather et al., 2025), and in machine learning, via linear classifications / "linear probes" (as is mentioned in the paper).

**Questions:**

not necessary for the paper – are there any connections to your "representation size" measure and "intrinsic dimensionality" (e.g. as in this paper: https://arxiv.org/abs/2012.13255)

---

> ### Author Response · Authors · 2025-11-17
> **Response to Reviewer VV2M**
>
> Thank you very much for your thoughtful and encouraging review. We appreciate your positive assessment of the paper’s motivation, framing, formalism, and experiments. Below, we address each of your comments in detail and clarify how our work relates to prior non-causal approaches.
>
> ## W1: Relation to prior non-causal / decoding-based work
>
> Thank you very much for raising this important point. In the [general response](https://openreview.net/forum?id=pODHH9DLeA&noteId=bEjqHnFAGm), we clarify how our contribution relates to prior decoding studies and other approaches that characterize representation through recoverable information. Briefly, our aim is not to reject causal perspective and to present non-causal perspectives as new. Rather, our aim is to reconcile the tension between the causal perspective and non-causal perspective. To this end, we formalize non-causal perspective in a computationally tractable way and to use this formalization to analyze the structure of representations. The readout representation framework and the representation size metric provide a principled way to examine these properties at the level of individual inputs, thereby complementing and extending existing decoding-based approaches. Please see the general response for the detailed explanation.
>
> ## Q1: Relation to intrinsic dimension
>
> Thank you for highlighting this insightful connection to the intrinsic dimension, which we are also interested in. Although this relationship is still speculative, we consider the following scenario to be plausible:
>
> 1. Empirical studies (e.g., Pope et al., 2021) suggest that the intrinsic dimensionality of natural data is substantially smaller than the ambient dimensionality of intermediate features. This mismatch naturally introduces redundancy in the representation.
> 2. When redundancy exists, models can operate effectively within a lower-dimensional subspace, which is consistent with observations on intrinsic dimension of parameter optimization (Aghajanyan et al., 2020).
> 3. The same redundancy also implies that a single input can be represented by multiple features, suggesting the readout representations of an input have a size larger than zero.
> 4. However, based on our experimental results on feature dimension and representation size (Fig. 4, right), we observe that near the final layer these two quantities do not necessarily correlate. We therefore interpret these results as suggesting that representation size and intrinsic dimension are related but not identical. Since representation size directly quantifies whether information can be recovered, we believe that it captures an aspect of the representation that goes a step beyond the intrinsic dimensionality of the data.
>
> We will revise the manuscript to more explicitly discuss this relationship, as it reinforces the interpretation of representation size and situates it within established findings on intrinsic dimensionality.
>
> ## References
>
> Aghajanyan, A., Gupta, S., & Zettlemoyer, L. (2021). Intrinsic Dimensionality Explains the Effectiveness of Language Model Fine-Tuning. In C. Zong, F. Xia, W. Li, & R. Navigli (Eds.), *Proceedings of the 59th Annual Meeting of the Association for Computational Linguistics and the 11th International Joint Conference on Natural Language Processing* (pp. 7319–7328). Association for Computational Linguistics. https://doi.org/10.18653/v1/2021.acl-long.568
>
> Pope, P., Zhu, C., Abdelkader, A., Goldblum, M., & Goldstein, T. (2021). The intrinsic dimension of images and its impact on learning. In *International Conference on Learning Representations*. https://openreview.net/forum?id=XJk19XzGq2J

---

> > ### Comment · Reviewer_VV2M · 2025-11-24
> >
> > Thank you for your response. I maintain my score.

---

### Author Response · Authors · 2025-11-17
**General response (1/2)**

We thank all reviewers for carefully reading our submission and for providing detailed and constructive feedback. Reviewers found the problem formulation and motivation clear and valuable (VV2M, zdre, Zuon) and regarded the overall framing and positioning within the relevant literature as appropriate (VV2M). Our central goal is to reconcile the tension between hierarchical abstraction and the retention of fine-grained information, which several reviewers noted as an important and broadly relevant question for machine learning, neuroscience, and the brain sciences more generally (VV2M, Zuon).

To address this goal, we define representations in terms of input information recoverable from features, introduce *readout representations* and their *representation size*, and apply this framework to convolutional and transformer-based models in both vision and language. Reviewers highlighted this formalization as interesting, worthwhile, and conceptually valuable (VV2M, izjS, Zuon), and found the methodology sound (zdre). They also viewed the experiments as clearly connecting representation size to the main question and as demonstrating potential generality across modalities (VV2M, izjS, zdre). Finally, reviewers noted that the observed relationship between representation size and model performance suggests potential practical utility for understanding or improving deep models (izjS).

We respond below to the concern that was commonly raised by multiple reviewers.

---

> ### Author Response · Authors · 2025-11-17
> **General response (2/2)**
>
> ## Response on the relation to prior non-causal / decoding-based work (to reviewers VV2M and Zuon)
>
> We appreciate the request from reviewers VV2M and Zuon to clarify how our contribution relates to the extensive literature that adopts non-causal or decoding-based perspectives on neural representation.
>
> First of all, we fully agree that many studies have adopted non-causal approaches across neuroscience, machine learning, and philosophy. In neuroscience, decoding studies aim to recover perceptual content from brain activity (e.g., Horikawa et al., 2013; Cheng et al., 2023; Kamitani et al., 2025). In machine learning, feature inversion (Mahendran & Vedaldi, 2015; Feather et al., 2015) and linear probing (Alain & Bengio, 2018) characterize representations by the information that can be read out from features. Informational and teleological theories in philosophy (Dretske, 1981; Millikan, 1989) likewise define representation by its informational or functional content rather than causal origin. We explicitly acknowledged these works in the Introduction and Related Work sections. Reviewers pointed out several works that considers the neural representation from philosophical perspective (Churchland & Sejnowski, 1990) and other examples that examine neural representation from recoverable information (Sucholutsky et al., 2023; Findling et al., 2025). Overall, as the reviewers point out, the field have accustomed to interpret neural representation from recoverable information. However, to our knowledge, no previous studies have formulated this custom of representation in non-causal and computable way. We will properly incorporate the attempts in existing research that the reviewers pointed out into the paper.
>
> Second, we do *not* aim to reject causal perspective, nor to claim that non-casual perspectives are themselves novel.  Rather, the motivation of this work is to resolve the tension between causal and non-causal perspectives as mentioned in Introduction: “how can a system designed for hierarchical abstraction, which supposedly discards details, simultaneously preserve the fine-grained information required for downstream functions?”, which was also highlighted by Reviewer zdre. The idea of hierarchical abstraction is widely accepted in the field, particularly in the studies that characterize representation from its causal origin (DiCarlo et al., 2012; Kriegeskorte, 2008). At the same time, non-causal studies have shown that fine-grained information is retained in higher-level processing both in the brain and in deep neural networks (Zhang & Sejnowski, 1999; Majima et al., 2017; Mahendran & Vedaldi, 2015). Moreover, our perceptual experience is full of fine-grained details, which we can use for downstream tasks (for example, visual perception not only includes object categories but also preserves small details such as surface texture and tiny stains on surfaces). We believe that both perspectives are valid as an explanation of the neural information processing, and are important. Our aim is to reconcile the two, not rejecting one of them.
>
> To resolve the tension, we propose readout representation, a computationally tractable non-causal formalization of representation, which builds on the studies mentioned above. This formulation differs from previous work by providing a rigorous treatment of misrepresentation in decoding studies and accommodating cases in which multiple neural features potentially represent a single piece of information, both of which were not fully addressed in previous works. This formalization also allows us to examine the size of such representations, providing a new lens to reconcile the tension. In this sense, our contribution is complementary to prior decoding studies and offers a useful framework for relating causal and non-causal views.
>
> We will revise the manuscript to make these points clearer and to better articulate how our framework builds on and extends existing non-causal approaches.
>
> ## Reference
>
> Churchland, P. S., & Sejnowski, T. J. (1990). Neural representation and neural computation. *Philosophical Perspectives*, *4*, 343–382. https://doi.org/10.2307/2214198
>
> Findling, C., Hubert, F., Acerbi, L., Benson, B., Benson, J., Birman, D., Bonacchi, N., Buchanan, E. K., Bruijns, S., Carandini, M., Catarino, J. A., Chapuis, G. A., Churchland, A. K., Dan, Y., Davatolhagh, F., DeWitt, E. E. J., Engel, T. A., Fabbri, M., Faulkner, M. A., … Pouget, A. (2025). Brain-wide representations of prior information in mouse decision-making. *Nature*, *645*(8079), 192–200. https://doi.org/10.1038/s41586-025-09226-1
>
> Sucholutsky, I., & Griffiths, T. (2023). Alignment with human representations supports robust few-shot learning. In A. Oh, T. Naumann, A. Globerson, K. Saenko, M. Hardt, & S. Levine (Eds.), *Advances in neural information processing systems* (Vol. 36, pp. 73464–73479). Curran Associates, Inc. https://doi.org/10.48550/arXiv.2301.11990

---

### Author Response · Authors · 2025-12-01
**Updates on the manuscript**

Thank you all for the valuable discussion and feedback. Your insights have been very helpful in improving our paper. We have updated the manuscript to incorporate all of your suggestions in this discussion period. Below is a summary of changes organized by reviewer:


### **Reviewer VV2M**

**Relation to prior non-causal / decoding-based work**

- Clarified that our intention is not to reject the causal-view but to reconcile the tension in **Section 2 Related work**.
- Clarified the novelty of our study to the prior decoding studies in Section 2 Related work. Cited papers you have suggested (Churchland and Sejnowski, 1990; Kriegeskorte et al., 2008; Sucholutsky et al., 2023; Feather et al., 2025) in **Section 2 Related work**.

**Relation to intrinsic dimension**

- Explained the connection between our findings and intrinsic dimension in **Section 4.3** **Interpretation of readout representations**. Cited the papers you have suggested (Aghajanyan et al., 2021).


### **Reviewer izjS**

**Potential applications of our framework and testing hypothesis**

- Clarified the potential applications for instance-level analysis, and mentioned the potential of testing hypothesis using the example you have suggested in **Section 5 Discussion**.

**Missing diagonal hashes in Figure 4**

- Made sure that the hashes are displayed, and improved the visibility of **Figure 4**.


### **Reviewer zdre**

**Use of DIP in vision experiments**

- Mentioned the results of the experiment without DIP in the main text in **Section 4.1 Feature inversion from perturbed features**.
- Added more detailed interpretation of the experiment without DIP to **Appendix A Limitations and Appendix F**. This includes the interpretation of the relatively sharp increase in the input distance to the small feature perturbation.
- Added the plot you have suggested as **Figure 28** in **Appendix F2. Representation size on pixel space**.


### **Reviewer Zuon**

**Relation to causal view and misclassification**

- Explained our intention is not to reject the causal-view but to reconcile the tension in **Section 2 Related work**.
- Added detailed explanation on the reason why causal view fails to handle misrepresentation in **Appendix B**. In the section, we clarified the difference between that the failure of causal view in handling misrepresentation is compatible with the fact that misrepresentation can still be causally evoked by neural activities or stimulus. Also, we added the link to the section at the beginning of **Section 3.2 Case studies**.
- Explained how to locate where misrepresentation arises in the brain in the example of misclassification in **3.2 Case studies**.

**Relation to prior decoding-based work**

- Clarified the novelty of our study to the prior decoding studies in **Section 2 Related work**. We cited the paper you have suggested (Findling et al., 2025).

**The dependence on the choice of readout procedure $\pi$**

- Manifested the dependency, and mentioned that the careful consideration is required to select $\pi$ in the formal definition in **Section 3.1 Definition of readout representation**.
- Added more detailed discussion in **Appendix A**.

**Relation to metamer study**

- Add more detailed discussion of the relationship of metamer studies and ours in **Section 2 Related work**.

**Dependence on the measure and equivalence relation**

- Explicitly mentioned that the feature space $H$ and signal space $S$ are endowed with an equivalence relations in **Section 3 Readout representation**.
- Mentioned the comparison of other metrics in the main text in **Section 4.1 Feature inversion from perturbed features**.

**Treatment of intrinsic and measurement noise**

- Added explanation in **Appendix C**.

---

### Author Response · Authors · 2025-12-03
**Summary of discussions for AC (1/2)**

We appreciate the hard work of ACs in these difficult circumstances. For your convenience, we have prepared a summary of the discussions we had with each reviewer.

### **Reviewer VV2M (8)**

**Strengths.** The reviewer found our research question very interesting, the framing and placement of literature impressive, our formalization very interesting, and experiments clear and interesting.

**Questions.** The reviewer questioned the novelty of our study relative to prior non-causal/decoding-based work. The reviewer also suggested a possible connection to intrinsic dimension, while noting this was not necessary for the paper.

**Our response.** We clarified that our aim is *not* to reject the causal view or claim non-causal perspectives are novel, but rather to *reconcile the tension* between causal and non-causal perspectives through a computationally tractable formalization of neural representation. We also discussed the connection to intrinsic dimensionality as suggested.

**Response from the reviewer.** The reviewer acknowledged our response and maintained the score.

**Updates to manuscript.** We updated the manuscript to clarify our novelty relative to previous works, and mentioned the suggested connection to intrinsic dimension.

### **Reviewer izjS (6)**

**Strength.** The reviewer found our framework interesting and worthwhile, noted that the empirical results show a promising direction for future work, and highlighted the generality of the method. The reviewer also noted that our choice of feature inversion as a readout method is a reasonable starting point, and that our instantiation of representation sizes is reasonable and practical.

**Questions.** The reviewer asked us to explain potential applications of our framework, proposed the potential of using our framework for testing hypotheses, asked for our interpretation of the results in Figure 4 (including why the gap between correct and missed classifications increases along the hierarchy), and pointed out the missing hatches in Figure 4.

**Our response.** We discussed instance-level analysis as a promising application of our framework. Our results show that representation sizes are related to the performance of the model on an instance, which enables per-instance analysis of neural representations, unlike existing methods (RSA, CKA) that characterize representations at the group level. We agreed that our framework has potential for testing hypotheses. We also provided our interpretation of the results in Figure 4, including an explanation for the increasing gap along the hierarchy.

**Response from the reviewer.** The reviewer did not provide a follow-up response.

**Updates to manuscript.** We updated the manuscript to clearly explain the applications, including the suggested application of testing hypotheses, and fixed Figure 4.

---

> ### Author Response · Authors · 2025-12-03
> **Summary of discussions for AC (2/2)**
>
> ### **Reviewer zdre (4 → “inclined to accept”)**
>
> **Strength.** The reviewer appreciated that we tackle an interesting problem and highlighted the methodology as sound.
>
> **Questions.** The reviewer asked us to clarify the interpretation of our results without using DIP, noting that the relatively sharp increase in pixel distance might question our claim of an image being represented by multiple features.
>
> **Our response.** First, we explained the intention of choosing DIP as a prior: DIP was chosen carefully as a weak prior, as it does not introduce any bias from training data. Second, we clarified that the results of the ablation study of DIP do not refute our main claim. Even without DIP, the reconstructed images still retain fine-grained information that is required by humans and artificial neural networks to solve downstream tasks. The sharp increase in pixel distance reflects high-frequency noise that is irrelevant to the actual information used for downstream tasks. Additionally, we created and shared an additional figure for an experiment that the reviewer requested. The results demonstrate that the use of DIP alone does not explain the broad readout representations we observed.
>
> **Reviewer’s response.** The reviewer agreed that the high-frequency noise observed in the ablation study is "irrelevant" to our claim, found the additional results "***very* encouraging**," and recognized the novelty of our work, saying “**get the novelty** […] **this is a very new way of thinking about things.**” The reviewer was "**inclined to raise**" their score. The reviewer also asked us to clarify the applications in neuroscience, which we answered after the reviewer engagement closed.
>
> **Updates on our manuscript.** The discussion helped us polish the interpretation of the ablation study of DIP, and we updated the manuscript to clearly explain them.
>
> ### **Reviewer Zuon (6 → 8)**
>
> **Strength.** The reviewer found the paper timely and our research question broadly important, and regarded the formalization as valuable.
>
> **Questions.** The reviewer questioned the novelty of our framework relative to prior decoding studies in neuroscience. The reviewer pointed out that the class of readout procedure $\pi$ was not adequately specified, and noted the implicit assumption of metric space on the input and feature spaces. The reviewer also asked us to clarify the treatment of intrinsic and measurement noise under our framework, the relation to misrepresentation, and the relation to metamer studies (Feather et al., 2023).
>
> **Our response.** We clarified the novelty of our framework relative to prior decoding studies. We explained how intrinsic and measurement noise are treated, and how misrepresentation is handled. We agreed with the reviewer on mentioning the importance of carefully choosing the readout procedure $\pi$, and the assumption of equivalence relation on the feature space $H$ and signal space $S$.
>
> **Reviewers response.** The reviewer mentioned that our "**clarifications are helpful**" and stated "**I'm satisfied that the paper is suitable for acceptance, and will raise my score.**" The reviewer raised the score to 8 before it was eventually reverted by the system.
>
> **Updates on our manuscript.** We updated our manuscript to accommodate the discussions. We added more careful explanation of the novelty relative to prior decoding studies, treatment of noise, and relation to misrepresentation. We also mentioned the importance of careful consideration in choosing $\pi$. Additionally, we made the equivalence relation on the feature space $H$ and signal space $S$ explicit, and discussed the relation to metamer studies.

---

### Meta-Review · Area_Chair_zHx4 · 2026-01-09

**Summary:**

#### Reviewer VV2M

The framing of the causal-only view is a strawman.

#### Reviewer izjS

The broader implications for the results are not clear.

#### Reviewer zdre

Like Reviewer izjS: the broader implications for the results are not clear.

#### Reviewer Zuon

(1) The concepts in the paper are claimed as novel, but are not in fact novel; (2) the paper lacks specificity on the implication of the choice of decoder for their claims.

**Reviewer Concerns:**

Reviewer VV2M's and Reviewer Zuon's (1) shared concern on the treatment of prior work in decoding was minor and addressed by discussion. The shared concern of Reviewer izjS and Reviewer zdre on the unclear implications for the results was at least partially addressed by discussion, if it was not conclusive. Lastly, Reviewer Zuon's (2) question about the significance of a technical detail was directly addressed by a proposed revision to the text.

**Reviewer Scores:**

I believe most reviewers would have upgraded their score to accept as all concerns were in the majority addressable, and so I can recommend acceptance.

---

### Decision · Program_Chairs · 2026-01-26

Accept (Poster)